# Fatty acid metabolism in neutrophils promotes lung damage and bacterial replication during tuberculosis

**Poornima Sankar[1], Ramon Bossardi Ramos[2], Jamie Corro[3], Lokesh K. Mishra[1], Tanvir Noor Nafiz[1], Gunapati Bhargavi[4], Mohd Saqib[1], Sibongiseni K. L. Poswayo[5], Suraj P. Parihar[5], Yi Cai[6], Selvakumar Subbian[4], Anil K. Ojha[3], Bibhuti B. Mishra[1] ***

1 Department of Immunology and Microbial Disease, Albany Medical College, Albany, New York, United States of America, 2 Department of Molecular and Cellular Physiology, Albany Medical College, Albany, New York, United States of America, 3 Division of Genetics, Wadsworth Center, New York State Department of Health, Albany, New York, United States of America, 4 Public Health Research Institute, New Jersey Medical School, Rutgers University, Newark, New Jersey, United States of America, 5 Center for Infectious Diseases Research in Africa (CIDRI-Africa) and Institute of Infectious Diseases and Molecular Medicine (IDM), Division of Medical Microbiology, Faculty of Health Sciences, University of Cape Town, Cape Town, South Africa, 6 Guangdong Key Laboratory of Regional Immunity and Diseases, Department of Pathogen Biology, Shenzhen University Medical School, Shenzhen, China

* mishrab@amc.edu

## Abstract

*Mycobacterium tuberculosis* (Mtb) infection induces a marked influx of neutrophils into the lungs, which intensifies the severity of tuberculosis (TB). The metabolic state of neutrophils significantly influences their functional response during inflammation and interaction with bacterial pathogens. However, the effect of Mtb infection on neutrophil metabolism and its consequent role in TB pathogenesis remain unclear. In this study, we examined the contribution of glycolysis and fatty acid metabolism on neutrophil responses to Mtb HN878 infection using *ex-vivo* assays and murine infection models. We discover that blocking glycolysis aggravates TB pathology, whereas inhibiting fatty acid oxidation (FAO) yields protective outcomes, including reduced weight loss, immunopathology, and bacterial burden in lung. Intriguingly, FAO inhibition preferentially disrupts the recruitment of a pathogen-permissive immature neutrophil population (Ly6G$^{lo/dim}$), known to accumulate during TB. Targeting carnitine palmitoyl transferase 1a (Cpt1a)-a crucial enzyme in mitochondrial β-oxidation-either through chemical or genetic methods impairs neutrophils' ability to migrate to infection sites while also enhancing their antimicrobial function. Our findings illuminate the critical influence of neutrophil immunometabolism in TB pathogenesis, suggesting that manipulating fatty acid metabolism presents a novel avenue for host-directed TB therapies by modulating neutrophil functions.

## Author summary

Tuberculosis (TB) caused by *Mycobacterium tuberculosis* (Mtb) is a significant global health issue. Neutrophils, immune cells that rapidly enter the lungs during Mtb infection,

**Data Availability Statement:** The RNA-seq datasets of mouse neutrophils have been deposited in the NCBI Gene Expression Omnibus (GEO) database with the accession number

GSE244230. Publicly available dataset used in this study to analyze expression of human genes includes GSE94438.

**Funding:** This study is supported by funding from National Heart, Lung, and Blood Institute (HL166257) and National Institutes of Allergy and Infectious Diseases (AI148239) of National Institutes of Health to BBM; AI32422 and AI163599 to AKO and JC. The funders had no role in study design, data collection and analysis, decision to publish, or preparation of the manuscript. The following authors received salary support from the NIH grants-BBM (HL166257, AI148239) and AKO (AI32422 and AI163599).

**Competing interests:** The authors have declared that no competing interests exist.

contribute both to fighting the infection and exacerbating lung damage. Their metabolic state, especially the way they produce energy, is crucial for their function during TB. However, the specific impact of Mtb on neutrophil metabolism and its role in TB progression has been unclear. In this study, we investigated the roles of glycolysis and fatty acid metabolism in neutrophil responses to Mtb infection. We found that inhibiting glycolysis worsens TB outcomes, while blocking fatty acid oxidation (FAO) provides protection, reducing lung damage, bacterial load, and overall disease severity. Notably, FAO inhibition specifically disrupts the accumulation of an immature neutrophil population that is more susceptible to Mtb, thereby improving infection control. Targeting carnitine palmitoyl transferase 1a (Cpt1a), a key enzyme in fatty acid metabolism, impaired neutrophil migration to infection sites and enhanced their antimicrobial activity. These findings suggest that modulating neutrophil metabolism, particularly through FAO inhibition, could be a promising strategy for developing host-directed TB therapies.

## Introduction

Neutrophils represent the most abundant type of white blood cells in humans and are fundamentally involved in the immune response against a myriad of pathogens [1,2]. Their significance becomes particularly evident in the context of *Mycobacterium tuberculosis* (Mtb), the causative agent of the human disease tuberculosis (TB) [3,4]. Recent studies have indicated an increased recruitment of these cells to the lung tissues to strongly correlate with the disease's severity [5–9]. The distinction between TB patients and healthy individuals, even those showing a positive reaction in tuberculin skin tests, is often marked by elevated neutrophil counts and a higher neutrophil-to-lymphocyte ratio [10]. Moreover, neutrophils are the dominant cell type in tissue biopsies of TB patients [11] and augmented neutrophil response has been closely linked to the lung damage associated with active disease [8,12]. This is further accentuated by the identification of neutrophil-specific type I IFN-inducible gene signatures that not only differentiate active TB cases from latent infections but also identify individuals at higher risk of progressing to active disease [13,14]. The predictive value of neutrophil counts regarding TB mortality risk [15] further reinforces the critical role these cells play in the immunopathogenesis of TB.

Despite their significance in pathogenic immune response to Mtb, the exact role of neutrophils in TB's clinical landscape remains debated. Initial recruitment of neutrophils to the site of Mtb infection and their antimicrobial actions, including the induction of oxidative damage to the bacteria, underline their importance in the innate immune response against TB [16,17]. However, the engagement of neutrophils in TB pathogenesis exhibits a spectrum of effects, from protective to potentially detrimental roles. This is vividly illustrated in animal models, where the presence of neutrophils correlates with disease severity and outcome in mice [18–20]. Furthermore, the impact of neutrophils on TB granuloma dynamics in nonhuman primates suggests a nuanced role that could either contribute to or mitigate pathogenesis, depending on the bacterial load and the stage of infection [21]. Therefore, a more in-depth investigation into the exact role of neutrophils in TB's complex immunological landscape is necessary.

Recent advances in immunometabolism have unveiled the critical importance of metabolic processes in immune cell function, adaptation, and survival, under both homeostatic and pathogenic conditions. Neutrophil metabolism, in particular, has emerged as a pivotal factor influencing their effector functions, lifespan, and the regulation of inflammation [22]. While

neutrophils are adept at eliminating bacteria through potent microbicidal mechanisms like reactive oxygen species (ROS) generation, their metabolic rewiring significantly impacts their functionality [23]. This involves a shift from aerobic respiration to glycolysis, a metabolic switch that fuels the production of ROS and other effector molecules [24]. While mature neutrophils accumulate glycogen reserves that serve as energy source at infection sites, especially in hypoxic conditions [25], immature neutrophils depend on β-oxidation of fatty acids for their ATP source [26,27]. The role of oxidative phosphorylation (OXPHOS) plays a negligible role in these inflammatory environments [28]. Interestingly fatty acid oxidation (FAO) plays a crucial role in neutrophil chemotaxis to infection sites in a bacterial pneumonia model [29]. Moreover, lipid metabolism in neutrophils and neutrophils like polymorphonuclear myeloid derived suppressor cells (PMN-MDSC) is essential for promoting metastatic cancer [30]. While significant metabolic plasticity has been reported in neutrophils in response to different tissue microenvironments [31], how Mtb infection shape the metabolic state of these cells remains to be explored.

The metabolic reprogramming of neutrophils in TB disease embodies a complex adaptive response to the pathogen and the microenvironmental cues within the host [32]. This reprogramming not only could affect neutrophil functionality but also impact the overall immune response to Mtb, influencing the disease outcome. However, the intricate details of how neutrophil metabolism is altered during TB and how these changes contribute to the pathogenesis of the disease represents a major knowledge gap which necessitates investigation into the intricate interplay between neutrophil metabolism, function, and their potential contribution to TB pathogenesis.

In this study, we employed a murine infection model with the hypervirulent Mtb strain HN878, a clinically relevant hypervirulent strain belonging to lineage 2 of W-Beijing isolates and utilized different mouse strains that represent diverse host microenvironments. We focused on the impact of glucose and fatty acid metabolism on neutrophil microbicidal function and the development of immunopathology during TB. Our findings reveal that inhibition of the FAO pathway in neutrophils significantly enhances the host's ability to restrict Mtb replication. This metabolic intervention not only improved infection outcomes but also had a profound impact on neutrophil recruitment and functionality, contributing to reduced immunopathology. Our research highlights the significance of neutrophil metabolism in the immune response to TB and establishes a foundation for deeper exploration into the immunometabolism of neutrophils. This understanding is crucial for uncovering key elements of the host-Mtb interactions, as well as for creating host-directed therapies against TB.

## Results

### Neutrophils are the major infected immune cells during Mtb HN878 infection

One of the defining features of TB in mice is the inflammation that damages the lung, leading to a decline in respiratory function and increase in mortality associated with immunopathology [33]. However, the cellular basis of the inflammatory environment in TB lung that is associated with disease susceptibility is poorly defined. We have previously reported that unchecked neutrophil influx to the lung is associated with susceptibility to the lineage 4 strain Mtb H37Rv infection [7,8,20]. To determine whether similar inflammatory environments develop during Mtb HN878 infection, we challenged mice with approximately 100 colony-forming units (CFU) of aerosolized Mtb (**Fig 1A**). We utilized a previously described single-strand DNA-binding protein SSB-GFP replication reporter in the Mtb HN878 background (*smyc*'::mCherry, SSB-GFP) to determine the replication status of Mtb in infected cells [34,35].

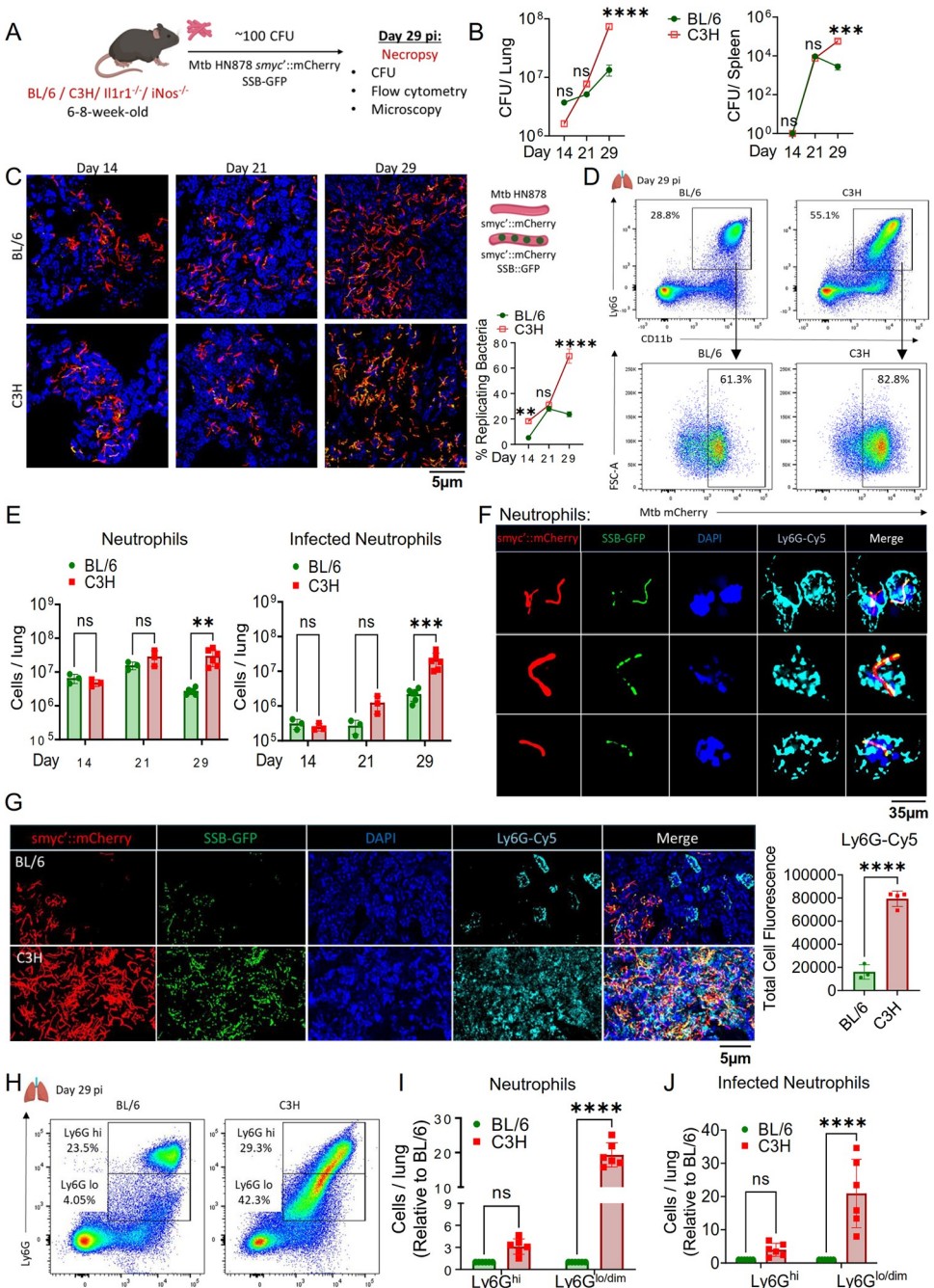

**Fig 1. Neutrophils are major infected cells in Mtb-susceptible mice. (A)** Experimental outline: Mice were infected with Mtb HN878 replication reporter and analyzed at indicated time points (days post infection- dpi). **(B)** CFU counts lung and spleen homogenates from BL/6 and C3H mice on 14, 21, and 29 dpi. **(C)** Representative confocal images of infected lung sections with DAPI staining, illustrating GFP+ replicating bacteria on mCherry+ bacilli and quantification of SSB-GFP foci. **(D)** Representative flow cytometry plots of neutrophils in lung tissue at 29 dpi. Top panel shows infiltrated neutrophils, gated from all live cells; bottom panel shows infected neutrophils, gated from all neutrophils. **(E)** Left panel: Enumeration of total live neutrophils (CD11b+ Ly6G+ Ly6C+ CD11c-) in infected BL/6 and C3H mice; Right panel: live infected neutrophils (Live CD11b+ Ly6G+ Ly6C+ CD11c- smyc'::mCherry+), both at 14, 21, and 29 dpi. **(F)** Confocal images of sorted lung neutrophils (CD11b+ Ly6G+) from C3H mice. Top, middle and bottom panels show three different neutrophils from three independent fields of view. Mtb is visualized by smyc':: mCherry (red), SSB-GFP (green), CD11b (cyan), with DAPI (blue). **(G)** Staining of lung sections with DAPI and Ly6G-Cy5 for analysis of bacterial replication in neutrophils via confocal microscopy. Total Ly6G-Cy5 fluorescence quantified. (H & I) Flow cytometric plots **(H)** and absolute count **(I)** of Ly6G-hi and Ly6G-lo/dim neutrophils (gated from

all live cells). **(J)** Infected Ly6G^hi and Ly6G^lo/dim neutrophils count in the lungs at 29 dpi. Data represent n = 3–6 mice per group, representative of 2 experiments. For confocal microscopy images (C, F), representative image of n = 3 mice / group; For (G) BL/6 n = 3 mice/ group, for C3H n = 4 mice/group; 3 fields of view per mouse; scale bar (C, G) 5μm, (F) 35μm; 200–500 bacterial rods per image were enumerated. Error bars depict Mean ± SD. Statistical analyses utilized two-way ANOVA with Tukey's multiple comparison test for (B, C, E, I, J) and an unpaired t-test for (G). *p<0.05, **p<0.01, ***p<0.001, ****p<0.0001; ns = non-significant. Illustrations created with www.BioRender.com.

In this reporter strain, the Mtb SSB protein is translationally fused to GFP, driven by the native *ssb* promoter, and Mtb undergoing active DNA replication exhibit green foci, providing a proxy for revealing the replication status of a given bacterium. Using this reporter strain, we infected both wild type C57BL/6 (BL/6) and C3HeB (C3H) mice, which develop lung pathology similar to human TB. The C3H mice showed increasing bacterial loads in their lungs and spleens over time, aligning with their known vulnerability to Mtb infection (**Fig 1B**). By day 29 post-infection (pi), about 75% of the bacteria in C3H mice lungs were replicating, compared to nearly 20% in BL/6 mice (**Fig 1C**). We then analyzed the cellular infiltration in their lungs to identify the leukocytes infected by Mtb. While the abundance of other leukocytes did not significantly differ between BL6 and C3H mice, neutrophil numbers surged 21 days pi. Moreover, these cells were the predominant infected cells in the lung of C3H mice though Mtb residence was also observed in monocytes and alveolar macrophages of these animals, mirroring human pulmonary TB, and highlighting the model's relevance (**Fig 1D and 1E**, **S1A–S1C Fig**). Further, we studied additional mouse strains previously shown to be susceptible to Mtb, specifically those lacking inducible nitric oxide synthase (*inos*^-/-^), and interleukin-1 receptor (*Il1r1*^-/-^), all within the BL/6 background. These mice also experienced a significant increase in lung neutrophils, similar to C3H mice (**S1D Fig**). Remarkably, the number of Mtb-infected neutrophils was significantly higher in these susceptible strains compared to BL/6 (**S1E Fig**). These findings indicate a clear link between neutrophils and intracellular Mtb growth with susceptibility to Mtb HN878 infection.

To assess bacterial replication within neutrophils, we sorted neutrophils (CD11b+ Ly6G+) from susceptible C3H mice and used confocal microscopy to analyze bacterial presence and replication in their cytospin samples (**Figs 1F and S2A**). We also sorted other immune cells from the lungs, including CD11b-negative cells, monocytes (CD11b+CD64+SigF-), and alveolar macrophages (CD11b+CD64^lo^SigF+), for comparison (**S2B and S2C Fig**). Among these, neutrophils were found to carry replicating Mtb (indicated by SSB-foci), while a very small fraction of alveolar macrophages also contained mCherry+ Mtb with SSB-GFP foci. No Mtb was detected in CD11b-negative cells or monocytes (**S2C Fig**). Additionally, we stained lung sections from both BL/6 and C3H mice at day 29 pi to examine the colocalization of the neutrophil marker Ly6G with SSB-GFP foci in mCherry+ bacilli. C3H lungs notably had more neutrophils and replicating bacteria colocalized with neutrophil marker Ly6G compared to the resistant BL/6 mice (**Fig 1G**). These findings further confirm that neutrophils facilitate bacterial replication in the lungs of susceptible mice.

We have previously reported a bacteria permissive neutrophil population, Ly6G^lo/dim in the lungs of susceptible mice following Mtb H37Rv infection [20]. Notably, these immature granulocytes accumulate in the lung as infection progresses and are more abundant in the lung of genetically susceptible mice. To examine if Mtb HN878 infection also recruits these neutrophils to the lung, we compared the neutrophils from BL/6 and C3H mice for their Ly6G expression. We observed an elevated number of Ly6G^lo/dim neutrophils in the C3H lungs at 29 days pi (**Fig 1H and 1I**). Notably, there was an increase in the infected Ly6G^lo/dim neutrophils (**Fig 1J**). These results support previous observations made in *inos*^-/-^ mice and further indicate that accumulation of Ly6G^lo/dim immature neutrophils provides a conducive cellular niche for Mtb replication.

To determine if this neutrophil-driven susceptibility as seen in mice is also relevant in other species, we employed a rabbit infection model of pulmonary Mtb infection, where disease outcome varies with the Mtb strain [36,37]. Infection with Mtb HN878 leads to progressive disease, severe inflammation, and lung pathology, including necrotic, caseating granulomas that can form cavities. Conversely, Mtb CDC1551 infection results in controlled infection and spontaneous latency establishment, following initial limited bacterial growth and mild pathology. The diverse pathological outcomes in rabbit lungs infected with these Mtb strains mirror the spectrum of TB seen in humans highlighting the usefulness of rabbit model for studying how neutrophil influx influences disease progression. Hence, we assessed neutrophil influx in rabbit lungs infected with Mtb HN878 or CDC1551 at 2- and 4-weeks post-infection, using comparative pathology. Similar to C3H mice, we found in rabbits infected with Mtb HN878 increased bacterial burden, necrotic lesions, more neutrophils, and Acid-fast bacilli (AFB) presence identified by Ziehl-Nielsen staining. Conversely, CDC1551-infected rabbits showed more lymphocytes and histiocytes in their lungs (**S2D–S2F Fig**), with nearly no detectable AFB in lung sections, indicative of a more controlled infection by this strain. These findings suggest Mtb HN878 infection triggers a significant neutrophil response that aid in bacterial growth in TB lesions. Our combined data from mouse and rabbit models of pulmonary TB underscore the role of neutrophils in promoting bacterial replication, which correlates with increased susceptibility to Mtb HN878 infection.

## Expression of metabolism associated genes are increased in neutrophils during TB

Since neutrophils from different host environments were not only quantitatively different, but they also varied in terms of their ability to harbor replicating Mtb, we sought to determine the transcriptome profile of these cells isolated from BL/6 and C3H mouse lungs. Neutrophils were extracted from the lungs of BL/6 and C3H mice following Mtb HN878 *smyc*'::mCherry, SSB-GFP infection, on day 29 pi and conducted bulk mRNA sequencing. Principal Component Analysis (PCA) unveiled significant transcriptomic differences among the lung neutrophils (**Fig 2A**). Through differential gene expression (DEG) analysis, we identified 452 upregulated and 1003 downregulated genes in C3H mouse lung neutrophils when compared to those from BL/6 mice (**Fig 2B**).

Subsequently, we performed gene ontology analysis of these DEGs to delineate the pathways that were up-and downregulated in lung neutrophils. Notably, we observed an upregulation in genes associated with metabolic pathways, including glycolysis, fatty acid transport, uptake, and catabolism in the neutrophils from C3H mice, compared to their resistant BL/6 counterparts (**Fig 2C**). Specifically, all genes essential for regulating glycolysis and β-oxidation of fatty acids exhibited increased expression in C3H mouse neutrophils from the infected mice (**Figs 2D and S3A**).

We then examined the metabolic gene signature in the previously published genome wide microarray dataset for the Mtb HN878 and CDC1551 infected rabbit lungs [36]. There was an overall increase in the expression of metabolism associated genes in the HN878 infected lung as early as 2 wks pi compared to CDC1551, linking these gene clusters in inflammatory disease and host susceptibility to TB (**S3B Fig**). Next, we analyzed the publicly available transcriptomics data of the peripheral blood cells (PBMCs) from a group of 98 TB patients and 314 healthy controls participated in a household contact study (GSE94438 dataset). This analysis revealed an elevated expression of genes including CPT1A, HADHA, HADHB, and ACAA2, implicated in the fatty acid oxidation (FAO) pathway. These genes mirrored the patterns observed in mouse lung neutrophils (**Fig 2E**). Other genes related to fatty acid and glucose

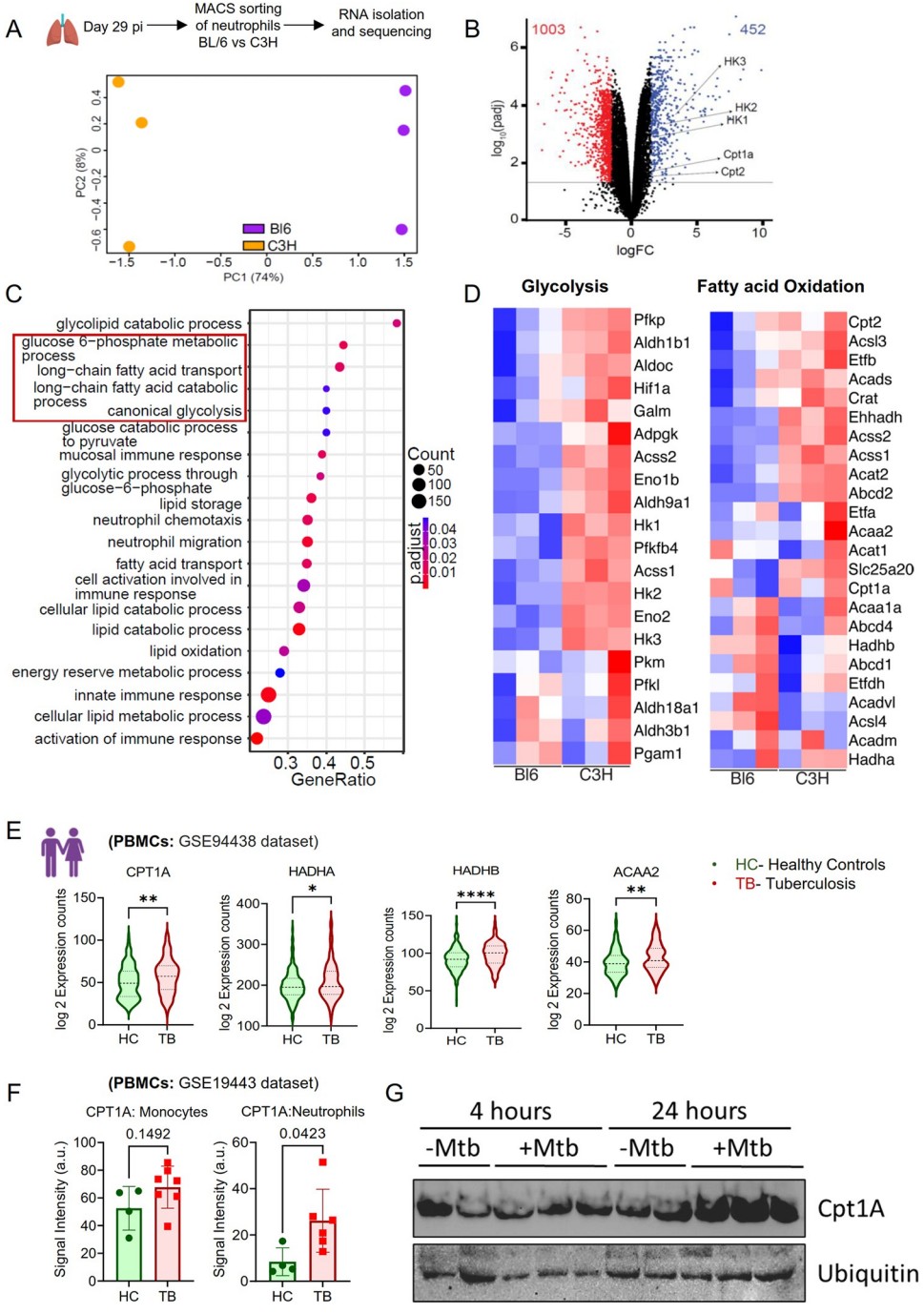

**Fig 2. Upregulation of metabolism-associated genes in neutrophils from TB-susceptible Mice.** Neutrophils were isolated from the lungs of BL/6 and C3H mice infected with approximately 100 CFU of Mtb HN878 replication reporter via aerosol and analyzed at 29 dpi. **(A)** PCA plot showing distinct gene expression profiles between neutrophils isolated from BL/6 and C3H mice, indicating divergent responses to Mtb infection. **(B)** Differential gene expression between resistant BL/6 and susceptible C3H mouse neutrophils is displayed, highlighting genes that are significantly upregulated or downregulated. **(C)** Graphs illustrating selected enriched pathways among the top 50 differentially expressed genes in neutrophils from BL/6 versus C3H mice, signifying potential metabolic differences in the immune response. **(D)** Heatmaps of Metabolic Pathways: Selected genes from glycolysis and fatty acid oxidation pathways are compared between BL/6 and C3H mouse neutrophils, with the heatmap color intensity representing expression levels. **(E)** Expression of genes implicated in Fatty acid metabolism in human PBMCs of healthy and pulmonary TB patients. RNA seq data from the public dataset reanalyzed as mentioned in methods. **(F)** CPT1A expression in monocytes and neutrophils from PBMCs of healthy and active TB patients. Data obtained by reanalyzing

the transcriptomics data from GSE19443 dataset. **(G)** Western Blot for Cpt1a (top) and Ubiquitin (bottom) from *ex-vivo* infected neutrophils at 4- and 24-hours pi. Each lane represents a biological replicate of a treatment condition, sourced from an independent well of a 12-well plate. sample size of n = 2–3 replicates per group, representative of two independent experiments. Error bars represent Mean ± SD for the indicated sample sizes (n = 3 mice/group for RNA sequencing. Statistical significance for gene expression differences in panel (E, F) was determined using an unpaired t-test, with the following significance markers: *$p < 0.05$, **$p < 0.01$, ***$p < 0.001$, ****$p < 0.0001$; ns indicates a non-significant difference. Clip art/Images within figure panels were created with www.BioRender.com.

metabolism were also altered in PBMCs of TB patients compared to healthy controls (**S3C and S3D Fig**). Further analysis of the microarray dataset from Berry et al [13] (GSE19443 dataset) revealed an upregulation of CPT1A gene that encodes for acetyl carnitine-palmitoyl transferase, a rate limiting enzyme in the mitochondrial FAO pathway in the blood neutrophils of human TB patients compared to healthy subjects (**Fig 2F**). Murine neutrophils infected with Mtb HN878 expressed higher amount of Cpt1a protein after 24hr pi (**Fig 2G**). Together, these cross-species metabolic gene signatures related to glucose and fatty acid metabolism suggest that metabolic programming during Mtb infection could be linked to TB pathogenesis.

To explore whether there were differences in glucose and fatty acid uptake during Mtb infection, we labeled both Mtb HN878 *smyc*'::mCherry infected and uninfected naïve bone marrow neutrophils with 2-Deoxy-2-[(7-nitro-2,1,3-benzoxadiazol-4-yl)amino]-D-glucose (2-NBDG), a fluorescent glucose analog and BODIPY FL-C$_{16}$ (4,4-Difluoro-5,7-Dimethyl-4-Bora-3a,4a-Diaza-*s*-Indacene-3-Hexadecanoic Acid (C16-BODIPY), a fluorescent fatty acid analog 4 hours prior to harvest. Subsequently, we assessed their uptake by flow cytometry as described previously [34]. Our results revealed that fatty acid uptake increased during Mtb infection while glucose uptake was impaired (**S4A–S4C Fig**). Next, we compared the glucose and fatty acid uptake efficiency of Mtb infected (Mtb+) and uninfected neutrophils (Mtb-) in the infected samples from S4B and S4C Fig. Although infected and uninfected neutrophils have comparable glucose uptake efficiency, Mtb-infected neutrophils significantly take up more fatty acids than Mtb-primed, yet uninfected neutrophils (**S4D and S4E Fig**). These findings indicate that fatty acid metabolism may be preferred over glucose for energy during neutrophil's response to Mtb HN878 infection.

## Fatty acid metabolism of neutrophils supports Mtb survival

Genes involved in the metabolic pathways of glucose and fatty acids exhibit significant upregulation in the neutrophils of Mtb-susceptible C3H mice, compared to their resistant BL/6 counterparts. Building on this observation, we sought to dissect the role of glycolysis and mitochondrial β-oxidation-in the neutrophilic response to Mtb. To this end, *ex-vivo* cultures of bone marrow-derived neutrophils were treated with specific inhibitors: 2-deoxy-D-glucose (2-DG), a glucose analog that impedes glycolysis, and a suite of inhibitors targeting fatty acid oxidation through mitochondrial β-oxidation. Among these, Etomoxir (ETO) acts as an irreversible inhibitor of carnitine palmitoyl-transferase 1a (*Cpt1*), thereby inhibiting FAO. Ranolazine (Ran), a clinical agent for angina, functions as a partial inhibitor of fatty acid oxidation (pFOX). Mildronate (Mil), an anti-ischemic drug, resembles γ-butyrobetaine, a precursor in carnitine biosynthesis, and it exerts its inhibitory effect on FAO by reducing plasma carnitine levels. Trimetazidine (TMZ), also indicated for angina, targets 3-Keto Acyl Thiolase (*Acaa*), a key enzyme in fatty acid β-oxidation, facilitating a metabolic shift towards glycolysis over FAO for energy needs (**Fig 3A**).

To assess the impact of Mtb infection on neutrophils, naïve bone marrow-derived neutrophils were isolated and infected *ex-vivo* with Mtb. 4-hour post infection (hpi), non-phagocytosed extracellular bacteria were removed, and the infected cells in culture the infected cells in

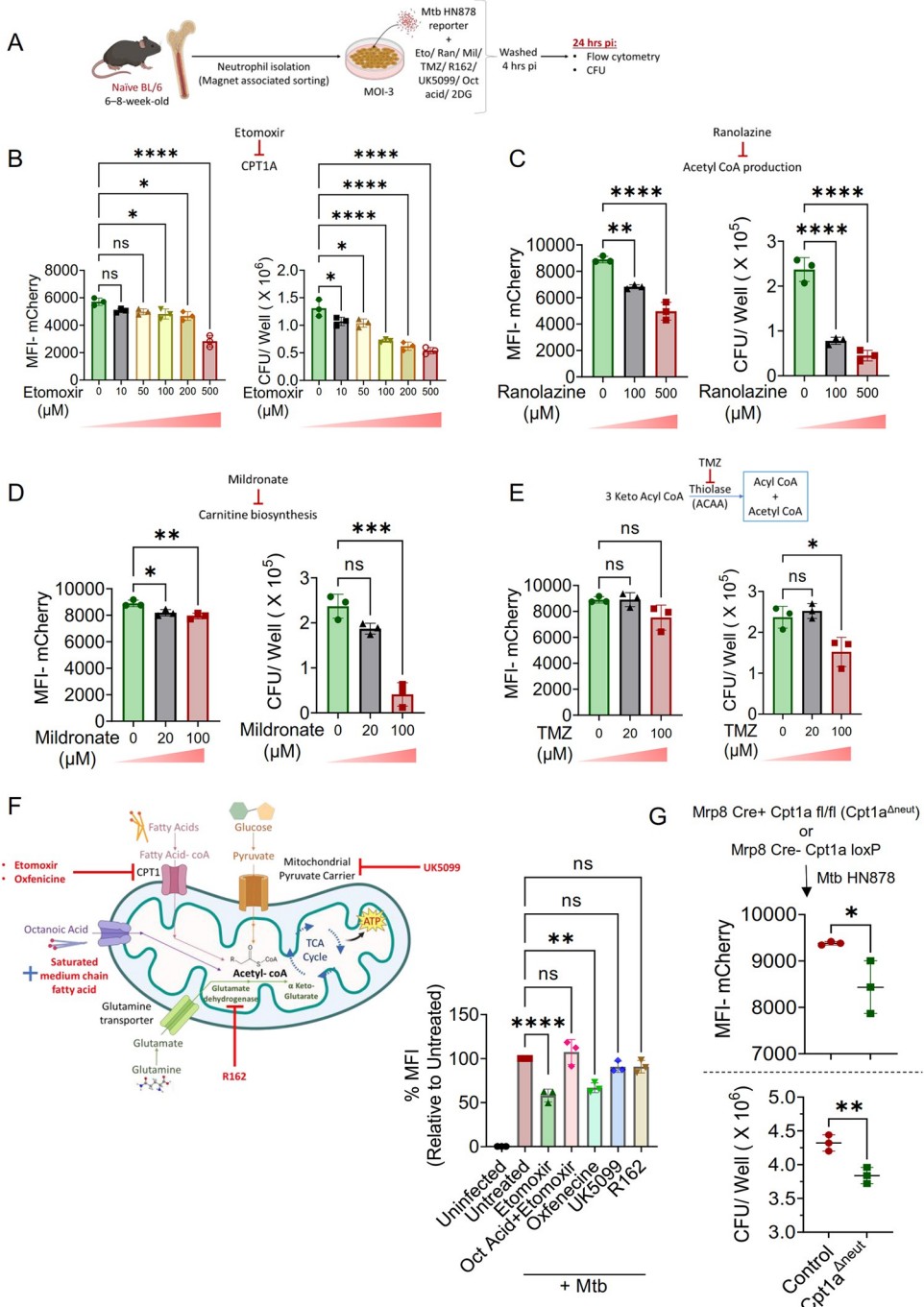

**Fig 3. Regulation of bacterial growth in neutrophils by mitochondrial fatty acid metabolism. (A)** Neutrophils from naïve BL/6 mouse bone marrows were isolated magnetically, infected with Mtb HN878 replication reporter at an MOI of 3, washed at 4 hours pi to remove extracellular bacteria, and incubated for another 20 hours. Neutrophil infection was then quantified using flow cytometry and CFU counts. **(B)** Effect of Etomoxir: Neutrophils were treated with increasing concentrations of Etomoxir. The MFI of Mtb HN878 *smyc*'::mCherry was assessed by flow cytometry (left panel), and the bacterial burden within neutrophils was measured by CFU counts at 24 hours pi (right panel). **(C-E)** Impact of Metabolic Inhibitors: Neutrophils were exposed to varying concentrations of Ranolazine **(C)**, Mildronate **(D)**, and Trimetazidine (TMZ) **(E)**. Subsequent MFI of Mtb-HN878-*smyc*'::mCherry was analyzed using flow cytometry at 24 hours pi (left panels), and bacterial burden was determined by CFU counts (right panels). **(F)** A schematic illustrates the mechanism of action of various small molecule inhibitors (left panel). Neutrophils were treated with 500μM Etomoxir, 5mM Oxfenicine, 50μM UK5099, 50μM R162, with additional assessments for the combined effect of 100μM Octanoic acid with 500μM Etomoxir. 24 hours pi, neutrophil infection levels were evaluated

by MFI of Mtb HN878 *smyc'*::mCherry, compared to untreated controls (right panel). **(G)** Genetic Model of FAO Deficiency: Mrp8 Cre+ Cpt1a fl/fl mice (Cpt1a$^{\Delta neut}$) with targeted FAO deficiency in neutrophils were generated. Neutrophils isolated from a Cpt1a$^{\Delta neut}$ mouse and a control littermate (Mrp8 Cre- Cpt1a loxP) were infected with Mtb HN878 reporter and analyzed for MFI of Mtb-HN878-mCherry at 24 hours pi by flow cytometry (top panel). Bacterial burden was also assessed by CFU counts at 24 hours pi in Cpt1a$^{\Delta neut}$ and control neutrophils (bottom panel). Sample size of n = 3 replicates per group, representative of two independent experiments. Error bars indicate Mean ± SD. Statistical analyses were conducted using ordinary one-way ANOVA for (B-F), and unpaired t-test for (G). Significance levels were calculated with Tukey's multiple comparison tests or unpaired t-tests, denoted by *p<0.05, **p<0.01, ***p<0.001, ****p<0.0001, ns for non-significant. Illustrations created with www.BioRender.com.

culture were analyzed after 24 hpi. The effects of glycolysis and fatty acid oxidation (FAO) inhibition on the microbicidal capacity of neutrophils were evaluated by quantifying the mean fluorescence intensity (MFI) of *smyc'*::mCherry via flow cytometry and enumerating colony-forming units (CFU) to measure intracellular bacterial load at 24 hpi. Given the stable half-life of the mCherry fluorophore and the short duration of our experimental design, we conducted a control experiment to confirm the robustness of MFI as a readout. Neutrophils in our *ex-vivo* infection model were treated with increasing concentrations of the Mtb bactericidal agent Isoniazid (INH), a first line antibiotic used for treating TB. As expected, this treatment resulted in a dose-dependent reduction in both the MFI of *smyc'*::mCherry and the CFU count in neutrophils (**S4F and S4G Fig**).

Inhibition of fatty acid metabolism using ETO, Ran, Mil, or TMZ led to a dose-dependent decrease in both MFI and bacterial CFU at 24 hpi (**Fig 3B–3E**). Importantly, the concentrations of these inhibitors did not exhibit cytotoxic effects on neutrophils, with the exception of 500 μM Ran (**S5A Fig**). Additionally, treatment with 2-DG significantly reduced the bacterial CFU within neutrophils at 24 hpi (**S5B Fig**). These findings underscore the critical role of glucose and fatty acid metabolic pathways in modulating the antimycobacterial function of neutrophils during Mtb infection.

To investigate whether neutrophils from an Mtb-susceptible background would exhibit similar effects upon FAO inhibition during *ex-vivo* infection, we infected C3H mouse neutrophils with our reporter Mtb strain. Similar to the results obtained with BL/6 mouse neutrophils, we observed reduced MFI of *smyc'*::mCherry with ETO, Ran, and Mil treatments, but not with TMZ or 2-DG (**S5C Fig**). Additionally, intracellular bacterial burden was reduced with ETO, Ran, Mil, TMZ, and 2-DG treatments (**S5D Fig**), further highlighting the importance of fatty acid metabolism pathways during Mtb infection.

To elucidate whether the reduction in bacterial load observed within neutrophils was attributable to the inhibition of specific metabolic pathways rather than a direct bactericidal effect of the inhibitors, Mtb was cultured in 7H9 broth supplemented with higher concentrations of the compounds for a 24-hour period, subsequent to which CFU counts were performed to assess bacterial viability. Treatments with ETO, Mil, and TMZ demonstrated no bactericidal activity against Mtb. In contrast, exposure to 5mM 2-DG and 500μM Ran exhibited toxicity towards the bacterial cultures. These findings suggest that the observed suppression of bacterial proliferation within neutrophils, at these heightened concentrations, may be attributed to the direct antimicrobial effects of 2-DG and Ran. However, the potential involvement of host metabolic pathways in mediating these effects remains a plausible contributing factor (**S5E Fig**).

## Mitochondria-dependent energy pathways in neutrophil response to Mtb infection

To explore the involvement of other mitochondria-dependent energy-generating mechanisms in the neutrophil response to Mtb, we targeted key enzymes in these metabolic pathways using

specific chemical inhibitors and evaluated their impact on intracellular bacterial proliferation at 24 hours pi (illustrated in the accompanying graphic). Mitochondrial pyruvate transport and amino acid metabolism were inhibited using UK5099 and R162, respectively. Given the documented non-specific effects of ETO beyond fatty acid oxidation, we employed Oxfenicine, an alternative inhibitor of Cpt1a, to assess whether it produced similar effects to ETO. Furthermore, we examined whether supplementation with fatty acids could counteract the effects of ETO. For this purpose, octanoic acid, a medium-chain fatty acid, was administered alongside ETO treatment (**Fig 3F**). Oxfenicine exhibited effects similar to those of ETO, notably causing a significant decrease in mCherry MFI, indicative of reduced bacterial load. The effect of ETO was reversed by co-treatment with octanoic acid, as shown by an increase in mCherry MFI. In contrast, treatment of neutrophils with UK5099 or R162 did not affect the bacterial load within these cells, suggesting that the observed effects on bacterial survival were specifically due to the disruption of fatty acid oxidation as a critical energy source (**Fig 3F**). Importantly, none of these inhibitors caused significant changes in neutrophil viability compared to untreated controls (**S5F Fig**). This specificity highlights the pivotal role of mitochondrial fatty acid metabolism in the neutrophil's response to Mtb infection.

Prior investigations have highlighted the significant role of fatty acid metabolism in the antibacterial response of macrophages, demonstrating that inhibiting this metabolic pathway can enhance the macrophage's capacity to eliminate H37Rv laboratory strain of Mtb [38,39]. To explore the potential analogous effects of ETO on macrophages during infection with Mtb strain HN878, bone marrow-derived macrophages (BMDMs) were infected with Mtb HN878 and subsequently treated with escalating doses of ETO. Aligning with the findings of preceding studies, a notable reduction in the MFI of *smyc*'::mCherry was observed at indicated time points pi (**S5G Fig**). This diminution in MFI correlated with a dose-dependent decrease in bacterial CFUs within ETO-treated BMDMs (**S5H Fig**). These findings provide substantial evidence supporting the hypothesis that Mtb infection-induced FAO could serve as a potential immune evasion strategy, enabling the pathogen to survive in neutrophils and macrophages.

To acquire genetic proof for the necessity of the mitochondrial FAO pathway in neutrophils, we generated a neutrophil-specific knockout of *Cpt1a* (Mrp8-cre Cpt1a[fl/fl], referred to as Cpt1a[Δneut]) by breeding Mrp8-cre[Tg] [40] with Cpt1a[loxP] mice [41] (**S6A–S6C Fig**). Analysis of Mtb infection in neutrophils isolated from these genetically modified animals demonstrated a significant reduction in both the MFI of *smyc*'::mCherry and the CFU counts, in comparison to their littermate controls (**Fig 3G**). These observations collectively reinforce the conclusion that FAO pathways in neutrophils are instrumental in supporting the intracellular survival of Mtb HN878.

## Inhibition of fatty acid oxidation ameliorated TB disease by reducing neutrophil infiltration

Our *in-vitro* findings have conclusively demonstrated the pivotal role of metabolism in neutrophil response to Mtb infection. Building on this, we explored whether inhibiting fatty acid metabolism enhances TB resistance in C3H mice, known for neutrophil-driven TB susceptibility (**Fig 1**). We employed the Mtb HN878 strain with a replication reporter to infect C3H mice. From day 17 to day 28 post-infection, we administered ETO (20 mg/kg), Ran (50 mg/kg), Mil (100 mg/kg), and TMZ (50 mg/kg) on alternate days by oral gavage. A separate cohort received 2-DG (250 mg/kg) to evaluate glycolytic inhibition's impact on TB outcomes. Disease parameters were assessed on 29 days pi as depicted in the schematic (**Fig 4A**). Treatment with fatty acid oxidation inhibitors significantly reduced weight loss, bacterial load in lungs and spleens (CFU counts), and improved lung pathology, evidenced by fewer inflammatory lesions

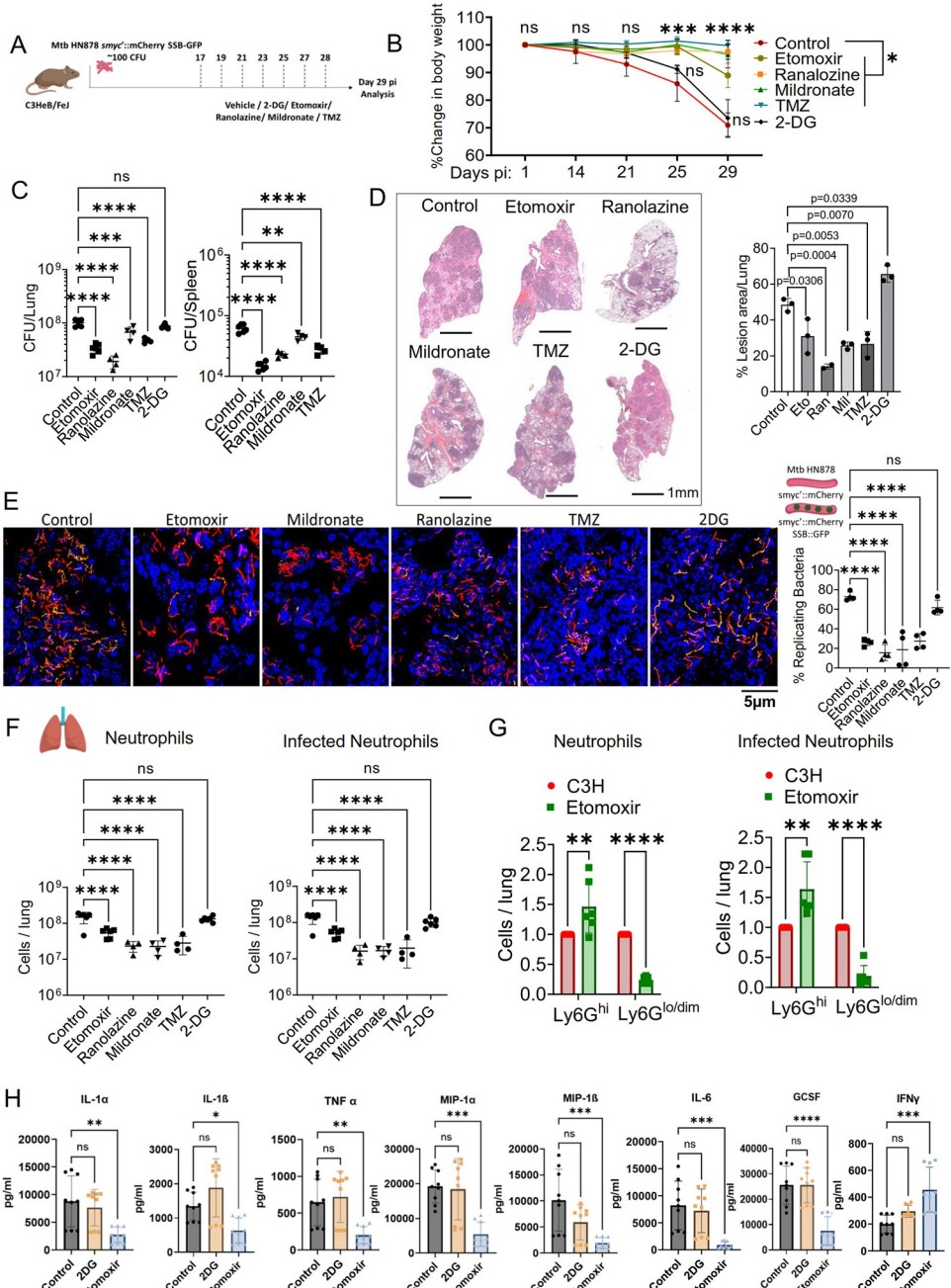

**Fig 4. Mitigation of TB disease severity through fatty acid oxidation inhibition *in-vivo*. (A)** Experimental outline: C3H mice were aerosol-infected with ~100 CFU of Mtb HN878 replication reporter and subsequently received alternate day treatments with metabolic inhibitors- 20 mg/kg ETO, 50 mg/kg Ran, 100mg/kg Mil, 50mg/kg TMZ, or 250 mg/kg 2-DG from day 17 to 28 by oral gavage. Assessments were conducted on 29 dpi. **(B)** Percentage of weight loss tracked in untreated and inhibitor-treated mice from day 14 to day 29 pi, with statistical comparisons at respective time points between inhibitor treated mice and untreated controls. **(C)** Bacterial load was determined in the lungs and spleens of untreated control and inhibitor-treated mice at 29 dpi by CFU counts. **(D)** Lung Histopathology: Images (left) and quantitative analysis (right; n = 3; For Ran, n = 2) of lesion areas within the lungs of both untreated controls and inhibitor-treated mice (Scale- 1mm). **(E)** Visualization and quantification of replicating bacteria (GFP+ on mCherry+ bacilli) in lung sections from infected untreated and inhibitor treated C3H mice at 29 dpi (Scale- 5μm). Representative image of n = 4 mice / group, 3 fields of view per mouse, 200–500 bacterial rods per image, from two experiments. **(F)** Flow cytometry assessment of live neutrophils (Live CD11b+ Ly6G+ Ly6C+ CD11c-) in the left panel and live infected neutrophils (Live CD11b+ Ly6G+ Ly6C+ CD11c- *smyc*'::mCherry+) in the right panel. **(G)** Neutrophil subset counts: Left panel illustrates total live Ly6G$^{hi}$ and Ly6G$^{lo/dim}$ neutrophil numbers in Etomoxir-

treated versus untreated mice. The right panel shows the count of live infected Ly6G$^{hi}$ and Ly6G$^{lo/dim}$ neutrophils in Etomoxir-treated animals compared to controls. **(H)** Cytokine profile: Levels of pro-inflammatory and protective cytokines measured in lung homogenates from untreated controls, and 2DG or Etomoxir-treated mice, including IL-1α, IL-1β, TNF-α, MIP-1α, MIP-1β, IL-6, G-CSF, and IFN-γ. Sample sizes ranged from n = 4 to 6 mice per group. Error bars represent Mean ± SD. Statistical analysis was conducted using ordinary one-way ANOVA for (C, D, E, F, H), and two-way ANOVA for (B, G), with significance determined by Tukey's multiple comparison tests: *p<0.05, **p<0.01, ***p<0.001, ****p<0.0001; ns denotes a non-significant result. Illustrations created with www.BioRender.com.

compared to controls (**Fig 4A–4D**). Contrarily, 2-DG treatment did not reduce weight loss or bacterial burden; it exacerbated lung inflammation, suggesting that global glycolysis inhibition might negatively affect host disease tolerance mechanisms (**Fig 4B–4D**).

We then utilized confocal microscopy to visualize lung sections from both untreated and inhibitor-treated mice, using DAPI counterstaining to identify cells harboring *smyc'*::mCherry bacilli and assess their replication status by quantifying SSB-GFP foci within the mCherry-positive bacilli. We analyzed the prevalence of bacteria across various fields of view, quantifying the proportion of bacteria exhibiting green foci (**Fig 4E**). This analysis confirmed that FAO inhibitors more effectively curtailed bacterial replication compared to both the vehicle-treated control group and the 2-DG-treated mice. Notably, the observed reduction in bacterial load and replication in FAO-inhibited samples was associated with a significant decrease in both the total neutrophil population and the number of infected neutrophils, compared to control and 2-DG-treated mice (**Figs 4F, S7A and S7B**). These findings further highlight the potential of FAO inhibition to limit Mtb-induced neutrophil influx, thereby constraining bacterial replication within these myeloid cells.

Building on our earlier findings, we further examined the differential counts of Ly6G$^{hi}$ (mature) and Ly6G$^{lo/dim}$ (immature) neutrophil populations in the treated mice, given the association of immature neutrophils with enhanced TB susceptibility, as detailed in Fig 1H–1J. ETO treatment resulted in a significant decrease in Ly6G$^{lo/dim}$ neutrophil accumulation and a concurrent reduction in their Mtb infection rates (**Fig 4G**). Conversely, ETO treatment led to an increase in mature Ly6G$^{hi}$ neutrophil infiltration compared to untreated controls (**Figs 4G and S8A**). Additionally, ETO administration reduced the infiltration of other immune cell types, including monocytes, macrophages, eosinophils, and dendritic cells (**S8B Fig**), with a marked decrease observed in the number of infected alveolar and interstitial macrophages (**S8C Fig**). Notably, ETO-treated mice also exhibited an elevated proportion of CD4+ and CD8+ T-lymphocytes (**S8D Fig**), suggesting that FAO blockade not only alters immune cell composition but also enhances protective immunity against TB by modulating infection dynamics within the lung.

To investigate the impact of ETO treatment on the inflammatory environment within the lungs of C3H mice, we quantified 32 cytokines, chemokines, and growth factors in lung homogenates, all of which play crucial roles in regulating inflammation and are particularly relevant to neutrophil-mediated pathologies. ETO treatment resulted in a significant decrease in pro-inflammatory cytokines, including IL-1α, IL-1β, and TNF, as well as a reduction in chemokines such as MIP-1α, MIP-1β, and MIP-2, which are essential for monocyte and neutrophil migration. Additionally, levels of G-CSF, a growth factor that promotes neutrophil survival, were markedly lower in ETO-treated mice compared to those receiving 2-DG or vehicle controls (**Figs 4H and S9**). In contrast, IFN-γ which is implicated in anti-mycobacterial host response [6,42] showed increased levels in ETO-treated lung homogenates, indicating the fostering of a protective host environment via fatty acid metabolism inhibition (**Fig 4H**). These findings support the notion that fatty acid metabolism plays a crucial role in driving

neutrophil recruitment to Mtb-infected pulmonary sites, facilitating bacterial replication in TB lesions-primarily within neutrophils—thereby exacerbating TB susceptibility.

## FAO inhibition improved TB outcomes in other susceptible mice

To assess whether the protective effect of FAO inhibition observed in C3H mice extends to other models of TB susceptibility, we used $Il1r1^{-/-}$ mice, which are highly susceptible to TB due to impaired IL-1 signaling. Given that early Mtb control in C3H mice is dependent on IL-1 [43,44], we treated $Il1r1^{-/-}$ mice with Mildronate (Mil, 100 mg/kg) every other day from 14 days pi (**Fig 5A**). Mil-treated $Il1r1^{-/-}$ mice exhibited significantly less body weight loss and a reduced lung bacterial burden compared to vehicle-treated controls (**Fig 5B and 5C**). Consistent with our previous findings, FAO inhibition in these mice led to a decrease in neutrophil influx to the lungs and a lower number of infected lung neutrophils (**Fig 5D and 5E**). There were no significant differences in the absolute numbers or Mtb-infection status of other immune cells, such as monocytes, macrophages, eosinophils, and dendritic cells (**Fig 5F and 5G**). Additionally, Mil treatment increased the frequency of CD3+ T-cells and CD19+ B-cells in the lungs (**Fig 5H and 5I**). These results suggest that FAO inhibition provides a broad protective effect against TB, even in the context of compromised IL-1 signaling, by modulating neutrophil dynamics and enhancing the adaptive immune response.

## Targeted inhibition of neutrophil FAO improves TB outcome

To delineate the specific role of neutrophil fatty acid metabolism in susceptibility to TB, we utilized conditional knockout mice we generated previously that specifically lack Cpt1a expression in neutrophils (Cpt1a$^{\Delta neut}$) and infected them with Mtb strain HN878 $smyc'$::mCherry via aerosol exposure. To simulate a model of increased susceptibility, we administered an anti-IL-1R monoclonal antibody (αIL1R) to both Cpt1a$^{\Delta neut}$ mutants and their littermate controls in a separate cohort (**Fig 6A**). Cpt1a$^{\Delta neut}$ mice demonstrated a significantly lower bacterial burden, reduced neutrophil infiltration, and fewer Mtb-infected neutrophils compared to control littermates. In these mice, the bacterial niche appeared to shift toward other myeloid cells, likely due to the reduced numbers of total and Mtb-infected neutrophils, which serve as a major myeloid niche for $Mtb$, even in the BL/6 background. Interestingly, in the Cpt1a$^{\Delta neut}$ mice with IL-1R1 inhibition in the lungs, we observed a reduction in inflammatory pathology, along with a marginal but statistically significant reduction in bacterial load compared to control littermates (**Fig 6A–6C**). This decrease in lung pathology and bacterial burden was correlated with an overall reduction in the total number of myeloid cells, including Mtb-infected neutrophils (**Fig 6C–6F**). While the reduction in other myeloid cells indicate a potential role of IL-1R signaling in regulating their infiltration, the reduction in neutrophils in the IL-1R inhibited lung was likely a result of Cpt1a inhibition as observed in both the wild type and IL-1R1 inhibited lungs. Additionally, we observed an increase in the frequency of CD3+T-cells and CD19 +B-cells in the lungs of untreated and IL1R-depleted Cpt1a$^{\Delta neut}$ mice, when compared to their respective control littermates following Mtb infection (**S10C and S10D Fig**). These findings underscore the detrimental role of Cpt1a-dependent mitochondrial fatty acid metabolism in neutrophils in modulating immune responses and controlling Mtb infection.

## Fatty acid metabolism is crucial for neutrophil migration

Prior research has implicated the inhibition of fatty acid metabolism, particularly through $Cpt1a$, in modulating neutrophil chemotaxis [29]. Our findings from both chemical and genetic modulation of fatty acid metabolism (Figs 4, 5, and 6) suggest that this energetic process plays a critical role in neutrophil recruitment to infection sites. Based on these

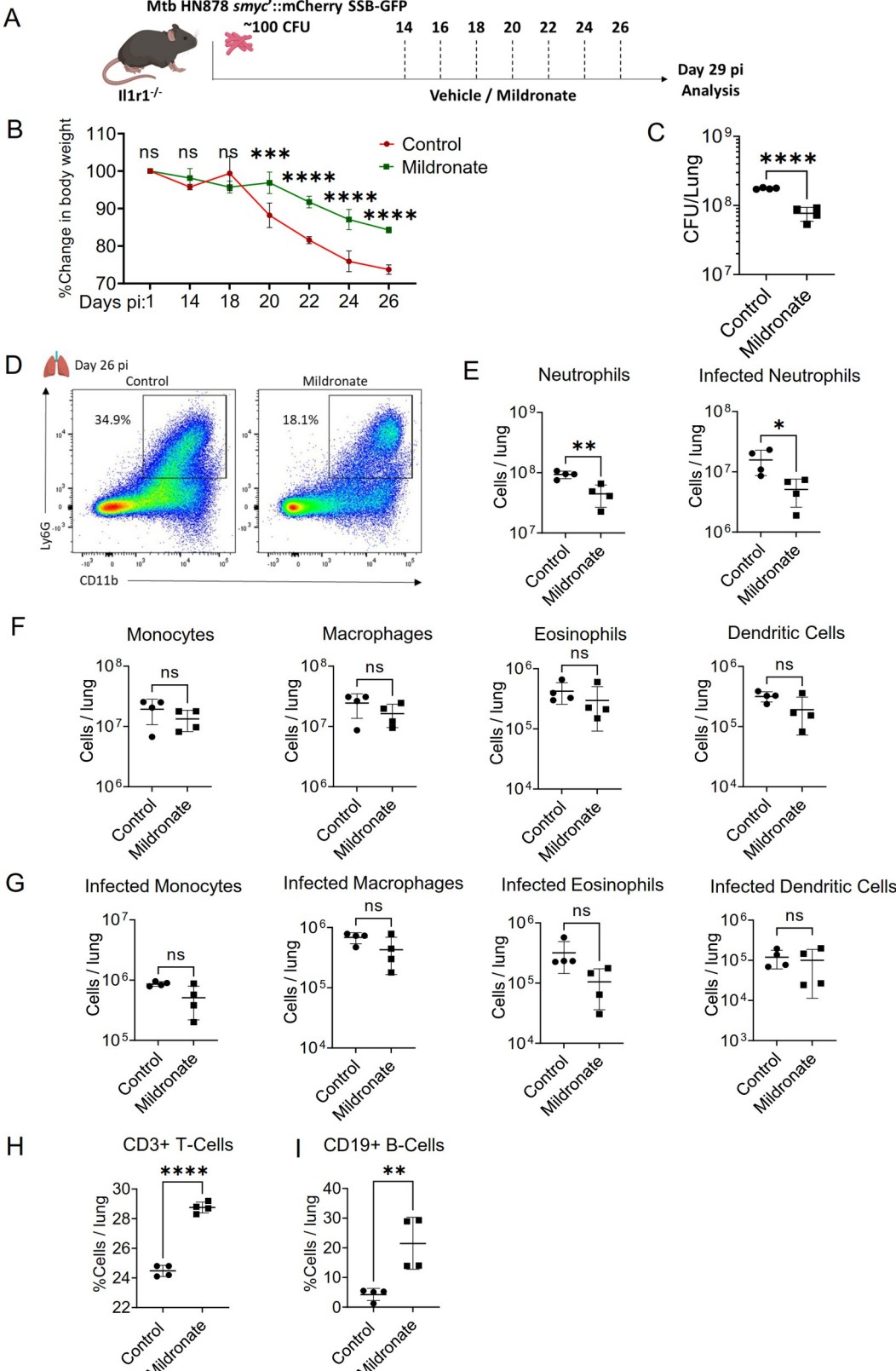

**Fig 5. Inhibition of fatty acid oxidation ameliorates TB disease in *Il1r1*−/− mice. (A)** Experimental overview: *Il1r1*−/− mice were infected with 100 CFU Mtb HN878 *smyc*::mCherry SSB-GFP via aerosol. Mice were either treated with 100mg/kg

Mildronate or vehicle for every alternate day by oral gavage from day 14 to 26 dpi, analyzed at 29 dpi. **(B)** Weight loss, tracked by percentage change in body weight in vehicle treated control mice and mildronate treated mice from day 14 to 26 days pi. **(C)** Bacterial burden in the lungs of control and mildronate treated mice, as enumerated by CFU counts ay day 29 pi. **(D)** Representative flow cytometric plot showing CD11b+ Ly6G+ neutrophils, gated from live cells from control and mildronate treated mice at 29 days pi. Quantification of total numbers of total neutrophils (Live CD11b+ Ly6G+ Ly6C + CD11c-) **(E)** and infected neutrophils (Live CD11b+ Ly6G+ Ly6C+ CD11c- *smyc'*::mCherry+) **(F)** from vehicle treated control and mildronate treated mice at 29 dpi. **(G)** Leukocyte counts: Enumeration of total number of monocytes (Live CD11b+ Ly6G- Ly6C+), macrophages (Live CD11b+ Ly6G- Ly6C- F4/80+), eosinophils (Live CD11b+ Ly6G- Ly6C- F4/80- SigF+), and dendritic cells (Live CD11b- Ly6G- Ly6C+ SigH+), in the lungs of control and mildronate treated mice at 29 dpi. **(H)** Total numbers of infected monocytes, infected macrophages, infected eosinophils and infected dendritic cells at 29 dpi. **(I)** Percentages of total CD3+ T-cells (left) and CD19+ B-cells in the lungs of control and mildronate treated mice at 29 dpi. Sample size, n = 4 mice/ group, from one experiment. Error bars indicate Mean ± SD. Statistical analyses were conducted using for unpaired t-test for (C, E-I) and two-way ANOVA for (B). Significance levels were calculated with Tukey's multiple comparison tests or unpaired t-tests, denoted by *p<0.05, **p<0.01, ***p<0.001, ****p<0.0001, ns for non-significant. Illustrations created with www.BioRender.com.

observations, we hypothesize that inhibition of FAO impairs neutrophil numbers in the lung by disrupting their chemotactic migration. To test this hypothesis, we conducted a transwell chemotaxis assay. Neutrophils treated with varying concentrations of etomoxir (ETO) were placed in the upper chamber, and their migration towards Mtb-infected murine bone marrow-derived macrophages (BMDMs), across a collagen-coated semi-permeable membrane, was measured after a 20-hour incubation. As a positive control, we used the well-known neutrophil chemoattractant N-Formyl methionine-leucyl-phenylalanine (fMLP) to validate the assay (**Fig 6G**).

As expected, uninfected neutrophils showed a dose-dependent increase in migration towards fMLP in the lower chamber. Similarly, robust migration of neutrophils was observed in response to Mtb-infected macrophages placed in the lower chamber. However, ETO treatment of neutrophils resulted in a dose-dependent inhibition of their chemotaxis towards Mtb-infected BMDMs, highlighting the significant role of fatty acid metabolism in neutrophil chemotaxis in response to Mtb infection (**Fig 6H**). Another FAO inhibitor, mildronate (Mil), also exerted inhibitory effects on neutrophil chemotaxis, though the effects were less pronounced compared to ETO (**Fig 6I**). Collectively, these results further underscore the critical involvement of fatty acid metabolism in regulating neutrophil responses during Mtb infection, suggesting that targeting this pathway could be pivotal in modulating immune responses to Mtb.

## Discussion

Despite recent advances in the field of immunometabolism, our understanding of neutrophil metabolism remains incomplete. Previous studies have indicated that neutrophils primarily rely on glucose as their primary carbon source for metabolic processes [22]. However, emerging evidence suggests that neutrophils exhibit metabolic plasticity as they can generate energy from a variety of carbon sources, including amino acids, carbohydrates, proteins, lipids, and the glycogen stores in their granules [23,24]. Neutrophils frequently encounter immunological environments with limited nutrient availability, necessitating their ability to adapt and employ diverse metabolic pathways to meet the demands of the immune response [24]. Our findings emphasize the crucial role of fatty acid metabolism in the two main effector functions to support intracellular bacterial replication and migrating to the lungs. However, it is worth noting that FAO blockade significantly blunted the influx of immature Ly6G^lo/dim neutrophils to the lung without altering the infiltration of mature Ly6G^hi neutrophils, suggesting that the immature pathogenic subset of these cells employ fatty acid metabolism for chemotaxis while mature neutrophils could derive energy from other metabolic pathways [26].

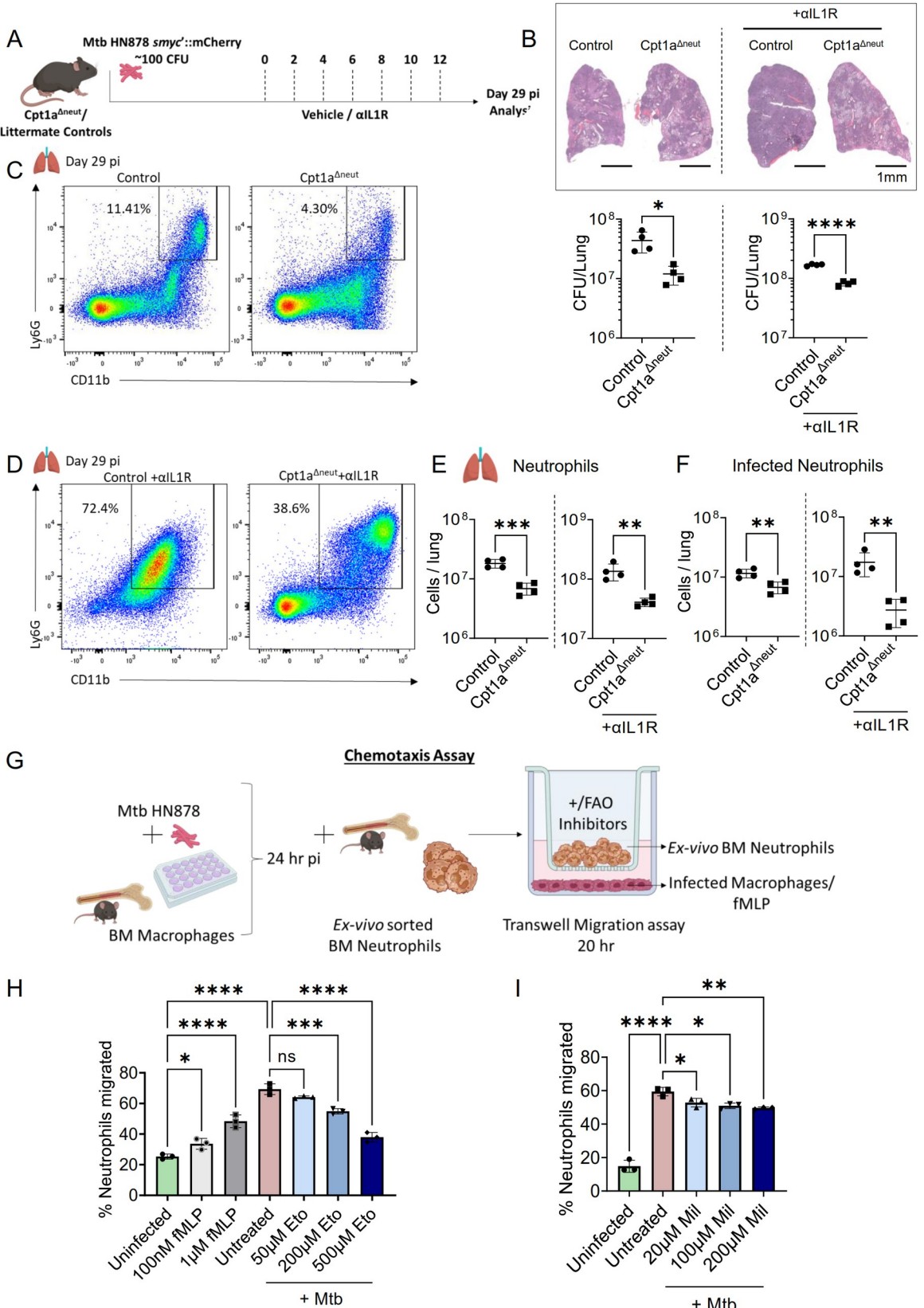

**Fig 6. Fatty acid metabolism in neutrophil migration to Mtb infection sites.** **(A)** Overview: Mrp8 Cre+ Cpt1$^{fl/fl}$ mice (Cpt1a$^{\Delta neut}$) and Littermate control mice (Mrp8 Cre- Cpt1a loxP) were infected with 100 CFU Mtb HN878 *smyc'*::mCherry by aerosol. In a separate cohort, both these groups of mice were treated with IL1R blocking antibody every alternate day intraperitonially from day 0–12 dpi and were analyzed at 29 dpi. **(B)** Representative lung histopathology images showing inflammatory lesions in Cpt1a$^{\Delta neut}$ and Control mice (top left) and Cpt1a$^{\Delta neut}$ and Control mice treated with α-IL1R antibody (top right); Scale- 1mm. CFU counts from lungs of control and Cpt1a$^{\Delta neut}$ mice (bottom left panel); α-IL1R treated control and Cpt1a$^{\Delta neut}$ mice at 29 dpi (bottom right panel). Flow cytometric profile showing CD11b+ Ly6G+ neutrophils (gated from live cells) from control and Cpt1a$^{\Delta neut}$ mice **(C)** and α-IL1R treated control and Cpt1a$^{\Delta neut}$ mice **(D)** at 29 dpi. Quantification of total numbers of neutrophils (Live CD11b+ Ly6G+ Ly6C+ CD11c-) **(E)** and infected neutrophils (Live CD11b+ Ly6G+ Ly6C+ CD11c- *smyc'*::mCherry+) **(F)** from control and Cpt1a$^{\Delta neut}$ mice (left) and α-IL1R treated control and Cpt1a$^{\Delta neut}$ mice (right) at 29 dpi. Sample size consisted of n = 4 mice per group (A-F), Data shown as Mean ± SD, are from one experiment. **(G)** Chemotaxis Assay Overview: Schematic representation of the transwell migration assay setup. Macrophages were infected with Mtb HN878 or treated with fMLP for 24 hours before neutrophils, treated with various concentrations of Etomoxir or Mildronate, were added to a collagen-coated transwell chamber to migrate towards infected macrophages for 20 hours. The extent of neutrophil chemotaxis was quantified and represented in a graph showing the percentage of migration into the bottom chamber. Graphs showing percentage of migrated neutrophils upon fMLP treatment/ Mtb treated/ Mtb with Etomoxir **(H)** and Mtb treated/ Mtb with Mildronate **(I)**. Representative of two independent experiments, n = 3 replicates per group. Error bars indicate Mean ± SD. Statistical analyses were conducted using unpaired t-test (B, E, F) and one-way ANOVA for (H, I). Significance levels were determined using unpaired t-test or Tukey's multiple comparison tests with the following notation: *p<0.05, **p<0.01, ***p<0.001, ****p<0.001; 'ns' indicates a non-significant result. Illustrations created with www.BioRender.com.

Recent studies have shed light on the link between Mtb pathogenesis and host metabolism [32]. Among the primary cellular niches for Mtb, macrophages play a central role. Within the lungs, alveolar macrophages (AMs) and interstitial macrophages (IMs) represent the major populations of infected macrophages, each exhibiting distinct metabolic profiles. AMs, which primarily rely on FAO, create a permissive environment for Mtb replication. In contrast, glycolytically active IMs limit infection [34]. Mtb enhances its reliance on mitochondrial oxidative metabolism, particularly exogenous fatty acids, and induces the formation of lipid-droplet-filled "foamy" macrophages [45]. These foamy macrophages are often found in the inner layers of TB lesions. Bacilli can be found in close proximity to intracellular lipid droplets, which are believed to serve as a source of nutrients in the form of cholesterol esters and fatty acids, creating a hospitable niche for the bacterium [8,46]. We and others have previously shown that neutrophils reside in close proximity to the caseum in the granulomatous lesions of susceptible mice including C3HeB [8,47], nonhuman primates [48] and humans [8,49] where they are exposed to the lipid-rich milieu [8,50]. Neutrophils could also serve as a source of these lipids in TB lesions as a consequence of cell necrosis induced by virulent mycobacteria [51]. However, the metabolic need of neutrophils in TB lesions has not been previously investigated. Our study provided compelling evidence for the involvement of mitochondrial β-oxidation of fatty acids in neutrophil response to Mtb. Similar to AMs, Mtb induces this metabolic pathway in neutrophils to grow within these cells. This is consistent with our previous report where we found, Mtb grow in neutrophil-rich lung lesions of C3H mice by overexpressing lipid metabolizing genes [8,52].

Oxidation of long-chain fatty acids requires their transport into the mitochondrial matrix, a process dependent on the carnitine palmitoyl-transferases (CPT) system. Cpt1, located in the mitochondrial outer membrane, catalyzes the conversion of an acyl-CoA into an acyl-carnitine, which is then passively transported through the outer membrane and traverses the inner mitochondrial membrane via the carnitine acyl-carnitine translocase. Cpt2 in the inner mitochondrial membrane converts the acyl-carnitine to acyl-CoA, which is then available for the iterative process of β-oxidation in the mitochondrial matrix. Cpt1 is considered the rate-limiting step in fatty acid oxidation. *Cpt2* deletion in myelomonocytic cells has been shown to enhance their antimycobacterial functions [38]. In our study, we showed that the genetic deletion or pharmacological inhibition of *Cpt1* in neutrophils (Mrp8-cre$^{Tg}$ Cpt1a$^{fl/fl}$ or Cpt1$^{\Delta neut}$) enhances their ability to suppress Mtb proliferation. Additionally, we found that inhibition of Cpt1 with ETO in BMDMs improves their capacity to control intracellular Mtb growth,

aligning with findings of Chandra et al [38]. The outcomes of treating mice with chemical inhibitors of FAO suggest that disease amelioration may result from the enhancement of antimicrobial functions in neutrophils and macrophages. In a parallel study, mice lacking Cpt1a specifically in neutrophils exhibited improved bacterial control compared to their littermates with wildtype Cpt1a in neutrophils and other cells. This enhanced bacterial control in Cpt1$^{\Delta neut}$ mice was associated with a reduction in inflammation relative to their wildtype counterparts (**Fig 6B**) These findings indicate that mitochondrial fatty acid metabolism may negatively influence the antimycobacterial immunity of myeloid cells. This could explain why lesions heavily infiltrated by myeloid cells, particularly neutrophils, tend to progress to an active disease state.

A pivotal finding from our research is the impact of FAO inhibition on the chemotaxis of neutrophils. The accumulation of neutrophils in the lungs is linked to tissue damage and the worsening of disease symptoms. Thus, controlling neutrophil recruitment to the lungs by targeting their fatty acid metabolism presents a promising therapeutic avenue. TMZ, a drug approved by the FDA for cardiovascular conditions such as angina, has been shown to possess nanomolar inhibitory effects on mitochondrial metabolism in macrophages [38]. Specifically, TMZ inhibits the activity of the long-chain 3-ketoacyl-CoA thiolase within the hydroxyacyl-coenzyme A (CoA) dehydrogenase trifunctional multienzyme complex subunit beta (HADHB), which is responsible for the final step in the β-oxidation of fatty acids. Echoing previous findings in macrophages, our study also demonstrates that TMZ treatment enhances the ability of neutrophils to inhibit Mtb growth, albeit at higher concentrations, likely due to the use of the hypervirulent Mtb HN878 strain. Additionally, the FAO inhibitor compounds tested in our study (TMZ, mildronate, and ranolazine) not only improve neutrophil-mediated control of Mtb growth in *ex-vivo* cultures, but also improved TB outcomes (reduced weight loss, bacterial load and mitigated immunopathology) in mice, suggesting these drugs could be repurposed for TB treatment. Such host-directed therapies (HDTs) are envisioned to complement existing antimicrobial treatments, potentially leading to shorter treatment durations.

The *in-vivo* efficacy of FAO inhibition through ETO in reducing Mtb replication, particularly within neutrophils, is notable. However, the broad impact of ETO at the dosages used in mouse models extends beyond neutrophils, affecting a variety of other cell types. This broad impact highlights the importance of further research into the consequences of FAO impairment across different cell types, especially within the myeloid and lymphoid lineages. Such studies will be crucial in uncovering new mechanisms of disease pathogenesis.

Previous research has noted off-target effects of ETO, particularly concerning the disruption of acetyl-CoA balance. Moreover, concentrations above 200 μM have been shown to impact IL-4 production from macrophages, influencing the M2 polarization of these cells. Despite these potential off-target effects, the dosage used in our *in-vivo* studies was selected for its non-toxicity and established safety, as reported in prior studies. Notably, our experiments did not reveal any significant changes in the levels of IL-4 and IL-13 cytokines in the lungs of mice treated with ETO, as indicated in **S9 Fig**. Nonetheless, the potential for other unanticipated off-target effects remains, warranting further exploration. Given these considerations, the application of ETO in the treatment of TB requires careful evaluation in future research. Despite these challenges, ETO's low cost and aqueous solubility make it a valuable research tool for investigating fatty acid metabolism's role in TB pathogenesis and potential treatment strategies.

In contrast to FAO, the inhibition of glycolysis by 2-DG had a moderate effect on bacterial replication but exhibited a significant exacerbation in tissue pathology (**Fig 4D**). 2-DG, an inhibitor of glucose uptake, completely inhibits glycolysis and affects the linked pentose phosphate pathway (PPP) and the TCA cycle, which utilizes pyruvate, an end-product of glycolysis,

as a substrate. Therefore, the observed increase in necrotic foci in the lung could be a result of the toxicity caused by the global inhibition of glycolysis in cell types that play a protective role during TB immunity. Indeed, a recent report has highlighted the protective role of glycolysis in myeloid cells for TB control [53]. These authors demonstrated that myeloid cell-specific deletion of lactate dehydrogenase subunit A (*Ldha*) enhances susceptibility to TB. Hence, the seemingly toxic effect of 2-DG treatment *in-vivo* could be a consequence of the loss of these protective effects. Therefore, an experimental approach that selectively targets *Ldha* in neutrophils would provide new insights into the role of glucose metabolism on their effector functions during TB.

The mechanisms by which FAO inhibition in neutrophils limits inflammation and enhances bacterial control remain to be fully elucidated. Our findings suggest that FAO inhibition affects neutrophil trafficking, as evidenced by the reduced number of neutrophils in the lungs following FAO blockade and the impaired chemotaxis of neutrophils towards (Mtb)-infected macrophages. This aligns with recent findings showing that mitochondrial metabolism and lipid uptake pathways are upregulated in activated lung neutrophils, which overexpress CD36, enabling increased lipid uptake and creating a permissive niche for *M. tuberculosis* (Mtb). Targeting these lipid uptake pathways has been shown to enhance the capacity of lung neutrophils to control Mtb replication [54]. Our study supports this observation and further demonstrates that inhibiting mitochondrial fatty acid metabolism can improve bacterial control by neutrophils and reduce TB pathology. Previous work from our group highlighted that neutrophil-rich TB lesions provide a nutrient-rich environment that supports bacterial replication by reducing oxidative and nitrosative stress, thus facilitating bacterial survival [8]. Our current study supports these findings, as evidenced by the predominance of replicative bacteria within these cells, highlighting the detrimental role of lipid metabolism of neutrophils in TB pathogenesis.

One significant finding that emerged from our study is the role of FAO-dependent energy in driving inflammation and disease in mice with defective IL-1 signaling. Given the protective function of IL-1 in early immunity to TB, and the development of type I interferon-mediated disease in hosts with defective IL-1 signaling, e.g., C3HeB mice and *Sp140*$^{-/-}$ mice in the BL/6 background that phenocopy the susceptibility of C3HeB mice [44,55], the results observed in IL-1R1-deficient mice treated with the FAO-blocking reagent, Mil, are particularly important (**Fig 5**). Furthermore, the observation that neutrophil-specific deletion of Cpt1a in mice (Cpt1$^{\Delta neut}$), when treated with anti-IL-1R mAb, led to reduced inflammation associated with fewer neutrophils and bacterial load compared to wildtype littermates, underscores a potential common mechanism of susceptibility. The impact of this genetic deletion on both bacterial control and the mitigation of immunopathology, while significant, is marginal compared to the effects observed in mice treated with chemical inhibitors (**Fig 4**). These findings suggest that, in addition to neutrophils, other myeloid cells-such as macrophages-utilize fatty acid metabolism to support bacterial replication in the lungs, thereby exacerbating inflammatory tissue damage. The ability of neutrophils to use fatty acid-mediated energy production to infiltrate the lungs and create a permissive niche for *M. tuberculosis* (Mtb) may represent a key mechanism in TB pathogenesis, particularly in individuals with increased susceptibility to the disease. Future studies should explore the metabolic requirements of neutrophils in other susceptible hosts, especially where IL-1-dependent and independent pathways contribute to pathogenesis.

However, our study has a few limitations. First, although we demonstrated the utilization of fatty acid oxidation by neutrophils in Mtb-infected lungs through genetic methods, we did not employ metabolomics-based analyses to confirm this. Additionally, we did not explore neutrophil-specific deletions of other metabolic enzymes involved in glycolysis (e.g., Ldha) or the

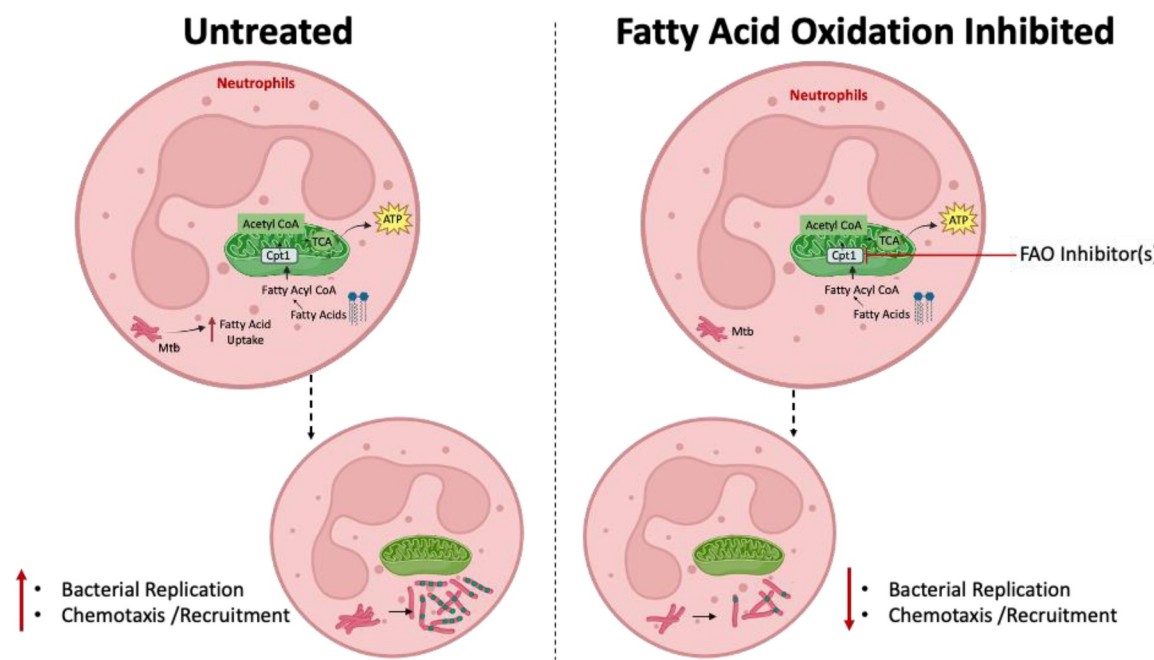

**Fig 7. Proposed model.** Neutrophils rely on the mitochondrial fatty acid oxidation pathway to meet their energy demands during Mtb infection. This bioenergetic process is essential for driving neutrophil migration to infection sites, where these immune cells inadvertently support bacterial growth, creating a replicative niche for the pathogen. Consequently, neutrophils utilizing fatty acids as an energy source act as a permissive environment for Mtb in the lungs of hosts susceptible to TB disease. Illustrations created with www. BioRender.com.

pentose phosphate pathway (e.g., Hk1, Hk2). Consequently, we were unable to determine whether neutrophils preferentially rely on fatty acid metabolism over glucose metabolism or if they exhibit metabolic plasticity in the inflammatory environments of susceptible hosts. Despite these limitations, our study elucidates the pivotal role of mitochondrial fatty acid metabolism in neutrophil responses to Mtb infection (**Fig 7**). This metabolic requirement is particularly evident in immature neutrophils, suggesting the potential for metabolic-based interventions in TB treatment strategies. Our investigation into neutrophil metabolism expands the realm of immunometabolism in infectious diseases, offering promising avenues for therapeutic innovation not only in TB but also in other infectious and inflammatory conditions.

## Materials and methods

### Ethics statement

All animal procedures adhered to the National Institutes of Health "Guide for the Care and Use of Laboratory Animals" standards. Animal protocols were approved by the Institutional Animal Care and Use Committee at Albany Medical College (ACUP #21–04003), in compliance with the Association for Assessment and Accreditation of Laboratory Animal Care, the US Department of Agriculture, and the US Public Health Service guidelines. All the rabbit studies were approved by the Institutional animal care and use committee (IACUC) of Rutgers University, NJ, USA.

### Mice

C57BL/6 (Stock #:000664), C3HeB/FeJ (Strain #:000658), $Il1r1^{-/-}$ (Stock #:003245), $iNos^{-/-}$ (Stock #:002596), B6. Cg-Tg (S100A8-cre,-EGFP)1Ilw/J (Stock#021614) and B6(Cg)-

$Cpt1a^{tm1c(EUCOMM)Hmgu}$/RjnsJ (Stock#032778) mice were purchased from The Jackson Laboratory. 6–8-week-old mice were used in this study. They were bred and maintained under Specific Pathogen Free conditions at Albany Medical College. All mouse studies were conducted in accordance with protocols approved by the AMC Institutional Animal Care and Use Committee (IACUC) (Animal Care User Protocol Number ACUP-21-04003). Care was taken to minimize pain and suffering in Mtb-infected mice.

## Bacterial strains

Hypervirulent *Mycobacterium tuberculosis* (Mtb) HN878 were used throughout this study. Mtb HN878 strains transformed with a plasmid encoding the fluorescence reporter *smyc'*:: mCherry, SSB-GFP (kindly provided by Dr. Sumin Tan, Tufts School of Medicine, Boston, MA), or with Mtb HN878 *smyc*::mCherry carrying Hygromycin B resistance. Bacteria were cultured in Middlebrook 7H9 media with OADC supplement, 0.05% Tween 20 and 0.5% v/v Glycerol and 50 μg/ml Hygromycin B in a shaking incubator at 37˚C for 5–7 days up to mid-log phase for bacterial infections. Strains were maintained in 20% Glycerol in -80˚C.

## Mouse infections

Single cell suspension of Mtb HN878 strains was prepared in phosphate buffer saline containing 0.05% Tween 80 (PBST). Clumps were dissociated by passing through 18 Gauge and 25 Gauge needles and ~100CFU bacteria were used for infection through aerosol route using a Glas-Col inhalation exposure system, Terre Haute, IN. Colony Forming Unit (CFU) was enumerated in serially diluted lung and spleen homogenates from infected mice at indicated days pi by plating on 7H10 Agar plates containing 0.5% v/v Glycerol and Middlebrook OADC enrichment. Colony counting was done in plates incubated at 37˚C after 3 weeks.

## *In-vivo* treatment of animals with metabolism inhibitors

C3HeB/FeJ mice infected with Mtb HN878 were treated with either 2-deoxy D-glucose (2DG) (250mg/kg) (Cat#111980250, Thermo Fisher) or Etomoxir (20mg/kg) (Cat# 11969- Cayman Chemical) or 50mg/kg Ranolazine (Cat# 15604, Cayman Chemical) or 100mg/kg Mildronate (Cat# HY-B1836, Med Chem Express) or 50mg/kg Trimetazidine (TMZ) (Cat# HY-B0968A, Med Chem Express) or PBS. Animals were treated every alternate day by oral gavage, starting from Day 17 to Day 28, for a total of 7 doses. IL1R$^{-/-}$ mice infected with Mtb HN878 were treated with either 100mg/kg Mildronate (Cat# HY-B1836, Med Chem Express) or PBS, every alternate day from Day 14 to Day 26 by oral gavage, for a total of 7 doses and analyzed at Day 29 pi. Cpt1a$^{Δneut}$ /Cpt1a$^{loxP}$ littermate control mice were treated with anti-mouse IL1R (CD121a) monoclonal antibody (BE0256-R025mg) every alternate day from Day 0 to Day 12 pi, intraperitoneally for a total of 7 doses. Percentage weight loss was calculated throughout the course of treatment. Mice were euthanized at day 29 pi and assessed for CFU count, histopathology, cytokines, and immune cells.

## Neutrophil isolation and *in-vitro* infection

Bones from naïve C57BL/6 / C3HeB/FeJ / Cpt1a$^{Δneut}$ /Cpt1a$^{loxP}$ littermate control mice were flushed with complete DMEM media (DMEM with Sodium Pyruvate, Sodium Bicarbonate, HEPES and 10% FBS). Extracted bone marrows were washed, passed through 18-gauge needles and then 40μM cell strainers to make single cell suspensions. Red blood cells were lysed by ACK lysis (Cat# BP10-548E, Lonza). From these cells, neutrophils were isolated by negative sorting by magnetic selection using the Mojo sort neutrophil isolation kit according to the

suggested protocol (Cat# 480058- BioLegend). In brief, single cell bone marrow suspension was washed with Mojo sort buffer (Cat# 480017, BioLegend) and incubated with the primary biotinylated antibody cocktail (1:10 in Mojo sort buffer) for 30 minutes. Following that, cells were washed and incubated in streptavidin bead bound secondary antibody (1:10 in Mojo sort buffer) for 30 minutes. Cells were then washed, and sorting was done over a magnet for 5 minutes (Cat# 480019, BioLegend). Unbound cells were collected, and magnet bound cells were discarded. Purity was checked by flow cytometry using CD11b (Clone M170) and Ly6G (Clone 1A8) surface staining. Purified neutrophils were counted and cultured in complete DMEM at 37˚C with 5% $CO_2$. For *in-vitro* infection, Mtb single cell suspension was prepared as above in complete DMEM and added onto neutrophils at the specified Multiplicity of Infection (MOI). 4 hours pi, cells were washed with fresh culture media to remove extracellular bacteria and they were analyzed by flow cytometry at 24 hours pi. For our experiments, we used 2 million sorted neutrophils per well in a 12 well non- treated cell culture plate and infected them with an MOI of 3.0 (6 X $10^6$ bacteria/ well) unless mentioned otherwise.

## Bone Marrow derived Macrophage generation and *in-vitro* infection

Naïve C57BL/6 mouse bones were flushed to isolate bone marrow cells. Single cell suspension of bone marrow cells was made as described above. Following ACK lysis, these cells were cultured for 3–5 days DMEM media containing L929 conditioning media, Sodium Pyruvate, Sodium Bicarbonate, HEPES and 10% FBS). When the bone marrow cells have differentiated into macrophages, the media was changed to complete DMEM and infected at an MOI of 3 with our Mtb HN878 reporter strain. For our experiments, we seeded 3 million bone marrow cells per well for macrophage development in a 6 well cell culture treated plate, and these were infected at MOI = 3.0 (9 X $10^6$ bacteria/ well). Cells were either plated after 4hrs or washed to remove extracellular bacteria and analyzed at 3- or 5-days pi by flow cytometry quantification of smyc':mCherry or lysed and plated for CFU conts as described below.

## Evaluation of bacterial burden by Colony Forming Units (CFU) counts

1. *In-vitro*: Infected neutrophils in culture were harvested 24 hours pi in PBST. Infected macrophages in culture were harvested at 3 days/ 5 days pi in PBST. The cell pellet was lysed with PBS+0.1% Triton X100 for 5 minutes at room temperature. Following that, serial dilutions were made in PBST and plated on 7H10 agar plates with 0.5% v/v Glycerol and OADC enrichment. Colonies were counted 3 weeks post incubation at 37˚C to evaluate bacterial burden in neutrophils pi.

2. *In-vivo*: Lung lobes and spleens from infected mice were homogenized using Matrix lysing tubes (Cat# 116913500, MP Bio). Tissues were homogenized in PBST, in a bead beater homogenizer. Serial dilutions were made in PBST and plated on 7H10 Agar plates containing 0.5% v/v Glycerol and OADC supplement. CFU were enumerated 3 weeks following incubation at 37˚C.

## *In-vitro* treatment of neutrophils

To test *in-vitro* uptake of fatty acids or glucose, 4 hours prior to harvesting infected neutrophils, 25µM C16 BODIPY (Cat# D3821- Invitrogen) or 15µM 2NBDG (Cat# N13195- Invitrogen) was added directly onto the cells. Cells were then stained and processed for flow cytometry. Neutrophils were either untreated or treated with indicated concentrations of 2-DG (Cat#111980250- Thermo Fisher); Etomoxir (Cat# 11969- Cayman Chemical); Ranolazine (Cat# 15604, Cayman Chemical); Mildronate (Cat# HY-B1836, Med Chem Express); Trimetazidine (Cat# HY-B0968A, Med Chem Express); Oxfenicine (Cat# 33698- Cayman

Chemical); UK5099 (Cat# 16980- Cayman chemical); R162 (Cat# 30922- Cayman Chemical); Octanoic Acid (Cat# C2875- Sigma Aldrich) with Etomoxir. Cells were processed as below for flow cytometry.

## Flow cytometry

Lung tissues were harvested from infected mice at stated time points and collected in cold PBS. These tissues were chopped and digested at 37°C in a Collagenase Type IV (150U/ml) (Cat 17104019, Gibco) and DNase I (60U/ml) (Cat# 10104159001, Roche- Sigma Aldrich) cocktail. Post digestion, the suspension was passed through 40μm cell strainers and ACK lysed to get single cell suspensions that were used for further staining. All subsequent steps were done in cells resuspended in FACS Buffer (PBS+0.5% BSA). For *in-vitro* experiments, cells were washed at indicated time points in FACS buffer. Fc-Block CD16/32 (Cat# 156604- BioLegend) was used to restrict non-specific antibody binding. Surface staining was done with directly conjugated antibodies at 4°C in the dark for 30 minutes. The cells were then fixed with Fixation Buffer (Cat# 420801- Bio legend) according to manufacturer's instructions and taken out of the BSL3 facility. All samples were analyzed on BD Symphony flow cytometer and all further analyses were done in FlowJo v10. For cell sorting, fixed cells were sorted on BD FACSAria II cell sorter and analyzed for sort on FACS Diva Version 6.1.3. Dead cells were excluded with eFluor 780 conjugated fixable viability dye (Cat# 65-0864-14 eBioscience) staining and further gating to analyze various populations was done on live cells. The antibodies used to analyze myeloid cells included: CD11b (Clone M170), Ly6G (Clone 1A8), Ly6C (Clone HK1.4), F4/80 (Clone BM8), CD11c (Clone N418), Siglec F (Clone S170072 or 1RMM44N), Siglec H (Clone 551), CD64 (Clone X54-5/7.1). Antibodies used to analyze lymphoid cells included: CD3 (Clone 145-2C11), CD4 (Clone GK 1.5), CD8 (Clone 53–6.7), CD19 (Clone 6D5). These antibodies were used in either APC or Alexa Fluor 647 or PE Cy7 or Alexa Fluor 700 or Brilliant Violet (BV) 450 or 421 or 510 or 605 or 711 or PerCP Cy5.5 in different panels. All antibodies were purchased from BioLegend.

## Immunofluorescence microscopy

Lung tissues from infected mice were fixed in 10% Buffer Formalin and were transferred sequentially to PBS, 15% Sucrose in PBS and 30% Sucrose in PBS and OCT embedded. These blocks were cut at 5 μm thickness and mounted on glass slides for further staining. For replicating bacteria visualization in lung sections, sections were washed for 5 minutes, twice with PBS-0.5% v/v Tween 20 (wash buffer) and once with PBS. The sections were then mounted with Prolong Gold Antifade reagent containing DAPI (Cat# P36935, Thermo Fisher). For neutrophil staining, sections were washed with wash buffer three times to rehydrate the slides. To minimize non-specific binding, the sections were blocked with PBS containing 4% BSA for 1.5 hours at room temperature. The sections were then washed thrice and permeabilized with freshly prepared PBS- 2% v/v Triton X-100 for 30 minutes at room temperature. Slides were then washed thrice and incubated for 12–16 hours at 4°C in a dark humidifying chamber with Alexa Fluor 647 Rat anti-mouse Ly6G antibody (Cat# 127609, BioLegend), prepared in PBS 0.5% BSA. Post incubation, the tissue sections were washed twice with wash buffer and once with PBS before mounting with Prolong Gold Antifade mounting medium containing DAPI counterstain overnight at room temperature. All slides were imaged on the Series 200 Dragonfly 901 series spinning desk confocal microscope (Oxford Instruments) with accompanying Fusion software. The images were captured on a 902 a Zyla sCMOS camera, and deconvoluted using Auto Quant (Media Cybernetics) using the Gibson-Lanni theoretical point-spread-

function (PSF). 3 fields of view per sample from 3–4 different mice per group (as mentioned in the figure legends) were acquired and analyzed on ImageJ.

### Immunofluorescence microscopy quantification

To quantify the percentage of replicating bacteria in lung sections, the total number of *smyc*'::mCherry+ bacilli were counted and the number of *smyc*'::mCherry+ rods with SSB-GFP foci as seen by GFP staining were enumerated and the percentage of mCherry positive bacteria with SSB-GFP foci is represented. 200–500 bacterial rods per image and 3 images from different fields of view from 3–4 biological replicates for each sample were enumerated (as mentioned in the figure legends). For neutrophil staining, the total cell fluorescence of Cy5 staining was calculated using Image J to assess Ly6G staining in lung sections. To this end, area and integrated cell density was measured from a minimum of 25 different regions of interest from each image. Similarly, the integrated cell density and area from a minimum of 5 different background regions (non-regions of interest) were measured for each image. The values were averaged to get a value of integrated density, selected area and mean fluorescence of regions of interest or background readings. The total corrected cell fluorescence for each field of view was calculated as the integrated density, subtracted from the product of total cell area selected and the mean fluorescence of background readings. 3–5 images were taken for each mouse lung section, and the total corrected cell fluorescence of all the images were averaged to generate one data point. 3–4 mice per group were analyzed.

### Cytospin of sorted cells for confocal microscopy

Sorted cells were counted and resuspended in PBS at a density of 0.5 X $10^6$ cells/ml. 200µl from this suspension was loaded into cytofunnels and centrifuged at 1000 rpm for 5 minutes in a Thermo Scientific Cytospin 4 Cytocentrifuge. The cells were then counter stained and mounted with Prolong Gold Antifade mounting medium containing DAPI (Cat# P36935, Thermo Fisher). The cytoslides were then imaged using Series 200 Dragonfly confocal microscope (Oxford Instruments), deconvoluted using Auto Quant (Media Cybernetics) and analyzed on ImageJ. 3 fields of view were taken from each of 3 mice/group. The data from 3 fields of view was averaged to get one data point.

### Histopathology

Formalin fixed infected lung tissues were paraffin embedded. 5 µm thick sections were cut and counterstained with hematoxylin and eosin (H&E) to analyze pathophysiology. H&E staining was performed at the histopathology core at Albany Medical College and were obtained on the NanoZoomer 2.0 RS Hamamatsu slide scanner and analyzed on Image J.

### Quantification of lesion area

H&E-stained lung sections were scanned using Nano Zoomer as mentioned above. Lesion areas were defined as regions of the lung occupied by inflammatory infiltration, while areas with minimal or no inflammation and open airways were categorized as uninvolved region. The total area of inflammation was then expressed as a percentage of the whole lung area. We used ImageJ software to select lesions, with the "Analyze" plug-in to quantify the area of each lesion, which was then summed to calculate the total area occupied by inflammatory lesions. Three mouse lungs were quantified in each group to assess statistical significance.

## Western blot

2 million neutrophils per well were infected with an MOI-3 of the reporter strain used for our *ex-vivo* model of infection as described above. Cells were either washed or processed for SDS-PAGE. 4- or 24-hours pi, cell samples were lysed in equal volume of 2X Laemmli sample buffer (Bio-Rad, USA) containing 0.5% β-Mercaptoethanol, boiled for 5 minutes at 95°C and then centrifuged at 4000-rpm for 10 seconds. Following this, samples were taken out of the BSL3 facility. To confirm our genetic knockouts, bone marrow neutrophils and other bone marrow leukocytes from Cpt1a$^{\Delta neut}$/Cpt1a$^{loxP}$ littermate control mice were sorted using magnetic sorting as described above. Total cell numbers from all mice were normalized and lysed and boiled as above. Equal volume (30 μL) of samples were loaded on 4–12% Mini-PROTEIN TGX Stain-Free Precast Gel (Bio-Rad) and the SDS-PAGE was run under reducing condition in 1X TGX buffer for 1-hour at 100 volts. The SDS-PAGE gel was transferred on nitrocellulose membrane (Trans-Blot Turbo Mini-size nitrocellulose) as instructed by the supplier, using semi-dry apparatus Trans-Blot Turbo transfer system. The protein bands were visualized by Ponceau S staining solution (Sigma, USA). The nitrocellulose membrane was blocked in 1.5% nonfat dry milk overnight at 4°C and subsequently incubated with rabbit anti-mouse monoclonal Cpt1a antibody (Cat# MA5-51291, Invitrogen) at a dilution of 1:250 dilution, overnight at 4°C. Thereafter, the nitrocellulose membrane was given three washes (10 minutes each) with 1X TBS buffer containing 0.05% Tween-20. The nitrocellulose membrane was incubated with goat anti-Rabbit IgG peroxidase conjugated secondary antibody (Cat# 31460, Invitrogen) at a dilution 1:5000 for 2-hours at RT and then washed three times with 1X TBST buffer containing 0.05% Tween-20. The nitrocellulose membrane was developed using chemiluminescent substate (Clarity Western ECL substrate, Bio-Rad). Following imaging, the membrane was stripped with Restore Western Blot Stripping Buffer (Thermo Fisher). For stripping, the membrane was washed with 1X TBS buffer and then incubated in Stripping Buffer for 15 minutes at RT with mild rocking. Thereafter, the membrane was blocked in 1.5% nonfat dry milk for 2-hours at RT. For Ubiquitin detection, primary antibody staining using Ubiquitin anti- mouse rabbit polyclonal antibody (IgG) (Cat# bs-1549R, Bioss) at 1:300 dilution at 4°C overnight, followed by anti-mouse IgG HRP-linked secondary antibody staining (Cat# 31460, Invitrogen) was done and imaged as above.

## Transwell chemotaxis assay

24 well transwell plates with 5 μm pore size were used for this assay. The semipermeable membrane layer was coated with 30 μg/ml collagen in 60% ethanol for at least 4 hours prior to the assay. These trans wells were calibrated with HBSS+ (with Ca+2 and Mg+2) by washing them twice with HBSS+ and leaving the coated plates in the 24 well plate with 1ml HBSS+ at 37°C for 1–2 hours. The lower chamber had BMDM cells, generated as mentioned above (1 X 10$^6$ cells/well) in complete DMEM, uninfected or infected with Mtb HN878 reporter at MOI-3, 24 hours prior to the assay, or with mentioned concentrations (100nM/ 1μM) of N-Formyl methionine-leucyl-phenylalanine (fMLP). 10$^6$ sorted neutrophils either pretreated for an hour with different concentrations of Etomoxir (50/ 200/ 500 μM) or Mildronate (20/100/200 μM) or untreated in 20 μl HBSS+, with or without Etomoxir/ Mildronate were added to the top chamber and allowed to migrate for 20 hours. Post migration, the top chambers were discarded. Migrated neutrophils along with trypsinized BMDM cells were collected, stained, and processed for flow cytometry.

## Cytokine analysis

Mouse lung homogenates were prepared in PBS at day 29 pi, and total protein concentrations were measured by Pierce BCA protein assay kit according to manufacturer's instructions.

Total protein concentrations were normalized in all samples and sent to Eve Technologies for quantification using the Luminex xMAP technology for multiplexed quantification of 32 Mouse cytokines, chemokines, and growth factors. The multiplexing analysis was performed using the Luminex 200 system (Luminex, Austin, TX, USA) by Eve Technologies Corp. (Calgary, Alberta). Thirty-two markers were simultaneously measured in the samples using Eve Technologies' Mouse Cytokine 32-Plex Discovery Assay (Millipore Sigma, Burlington, Massachusetts, USA) according to the manufacturer's protocol. The 32-plex consisted of Eotaxin, G-CSF, GM-CSF, IFNγ, IL-1α, IL-1β, IL-2, IL-3, IL-4, IL-5, IL-6, IL-7, IL-9, IL-10, IL-12(p40), IL-12(p70), IL-13, IL-15, IL-17, IP-10, KC, LIF, LIX, MCP-1, M-CSF, MIG, MIP-1α, MIP-1β, MIP-2, RANTES, TNFα, and VEGF. Assay sensitivities of these markers range from 0.3–30.6 pg/mL for the 32-plex. Individual analyte sensitivity values are available in the Millipore Sigma MILLIPLEX MAP protocol.

### Human patients whole blood transcriptome study

A publicly human gene expression dataset (GSE94438), comprising 412 samples (HC = 314, TB = 98) was obtained from GEO platform. The RNA sequencing data obtained for this study exclusively comprised unstandardized count data. Raw counts were normalized by variance stabilizing transformation (VST) in DESeq2.

Berry et al, 2010 (UK cohort): We accessed the publicly available blood transcriptome dataset (GSE19443) from Berry et al, 2010 [13]. Using the transcriptomics data, we determined the mRNA expression levels of CPT1A in monocytes and neutrophils from healthy donors and patients with active TB. The mRNA expression was expressed as signal intensity (arbitrary units) for each donor and was plotted and reanalyzed for statistical significance using a student t-test comparing two groups (Healthy controls versus active TB) in GraphPad Prism.

### Rabbit model of Mtb Infection, Histopathology, AFB staining and Transcriptome analysis

1. Mtb infection: New Zealand White rabbits (specific pathogen-free) of about 2.5 kg (Envigo, USA) were infected with aerosolized Mtb HN878 or Mtb CDC1551 strains using the "nose-only" exposure system (CH Technologies Inc., NJ, USA) to implant ~3.2–3.5 $\log_{10}$ CFU in the lungs (T = 0) as previously described [36,56]. At defined time points, rabbits (n = 4 per time point) were euthanized, and lungs were used for histopathologic examination and RNA isolation. All animal procedures were conducted according to the protocols approved by the Rutgers University IACUC (Institutional Animal Care and Use Committee).

2. Rabbit lung histology: Portions of rabbit lungs were randomly cut and fixed in 10% neutral formalin, followed by paraffin-embedding. The formalin-fixed and paraffin-embedded tissue blocks were cut into 5-micron sections and used for staining with hematoxylin & eosin (H&E) or acid-fast staining by Ziehl-Nielsen method (for Mtb). Stained lung sections were analyzed microscopically using Nikon Microphot-FX system with NIS-elements F3.0 software (Nikon Instruments, NY).

3. RNA isolation from rabbit lungs: After the autopsy, random portions of rabbit lungs with or without Mtb infection were immediately treated with TRI Reagent (Molecular Research Center, Cincinnati, OH) and total RNA was isolated, as described previously [37]. RNA was purified using RNeasy kit (Qiagen, CA, USA) and the quality and quantity were measured with Agilent 2100 Bioanalyzer (Agilent Technologies, CA, USA).

4. Genome wide transcriptome analysis of Mtb-infected rabbit lungs: Total RNA from rabbit lungs with or without Mtb infection was subjected to genome wide transcriptome profiling using 4X44k rabbit whole genome microarrays (Agilent Technologies, Santa Clara, CA), as described [57]. Briefly, arrays were hybridized with a mixture of Cy3 or Cy5 labeled cDNA, generated using uninfected or Mtb-infected rabbit lung RNA at each time point. The arrays were washed, scanned and data extracted. Background-corrected, normalized data were analyzed by One way ANOVA using Partek Genomics Suite Ver 6.8 (Partek Inc., St. Louis, MO); an unadjusted p value < 0.05 was used to select significantly differentially expressed genes (SDEG). The list of SDEG was uploaded into Ingenuity Pathway Analysis portal (IPA; Qiagen, CA) as described previously [37], to identify gene networks and pathways impacted by the SDEGs. The microarray data is deposited to Gene Expression Omnibus (accession numbers GSE33094 and GSE39219).

## RNA library preparation and sequencing of murine neutrophils

Single cell suspensions from Mtb infected mouse lungs were prepared as described above. Mouse neutrophils were sorted by positive selection using anti- Ly6G-APC for 30 minutes at 4˚C in the dark followed by secondary staining with anti-APC nanobeads following the APC nanobead kit protocol (Cat# 480090-Bio Legend). Purity was assessed using an anti-mouse GR1 antibody. These Ly6G+ cells were used for RNA isolation. Sorted Ly6G+ cells were stored at -80˚C in RLT Plus Buffer (Cat# 1053393-Qiagen). Total RNA isolation was done using RNeasy Plus Mini Kit (Cat# 74134- Qiagen) following the manufacturer's instructions. RNA samples were quantified using Qubit 2.0 Fluorometer (Thermo Fisher Scientific, Waltham, MA, USA) and RNA integrity was checked with 4200 TapeStation (Agilent Technologies, Palo Alto, CA, USA). rRNA depletion sequencing library was prepared by using QIAGEN FastSelect rRNA HMR Kit (Qiagen, Hilden, Germany). RNA sequencing library preparation uses NEBNext Ultra II RNA Library Preparation Kit for Illumina by following the manufacturer's recommendations (NEB, Ipswich, MA, USA). Briefly, enriched RNAs are fragmented for 15 minutes at 94˚C. First strand and second strand cDNA are subsequently synthesized. cDNA fragments are end repaired and adenylated at 3'ends, and universal adapters are ligated to cDNA fragments, followed by index addition and library enrichment with limited cycle PCR. Sequencing libraries were validated using the Agilent Tapestation 4200 (Agilent Technologies, Palo Alto, CA, USA), and quantified using Qubit 2.0 Fluorometer (ThermoFisher Scientific, Waltham, MA, USA) as well as by quantitative PCR (KAPA Biosystems, Wilmington, MA, USA). The sequencing libraries were multiplexed and clustered on one flowcell lane. After clustering, the flowcell was loaded on the Illumina HiSeq instrument according to manufacturer's instructions. The samples were sequenced using a 2x150 Pair-End (PE) configuration. Raw sequence data (.bcl files) generated from Illumina HiSeq was converted into fastq files and de-multiplexed using Illumina bcl2fastq program version 2.20. One mismatch was allowed for index sequence identification.

## RNA sequencing data analysis of murine neutrophils

After demultiplexing, sequence data was checked for overall quality and yield. Then, raw sequence reads were trimmed to remove possible adapter sequences and nucleotides with poor quality using Trimmomatic v.0.36. The reads were then mapped to the Mus musculus reference genome available on ENSEMBL using Rsubread v1.5.3. Gene counts were quantified by Entrez Gene IDs using featureCounts and Rsubread's built-in annotation. Gene symbols were provided by NCBI gene annotation. Genes with count-per-million above 0.5 in at least 3

samples were kept in the analysis. Differential expression analysis was performed using limma-voom.

Reads were mapped to Mus musculus genome available on ENSEMBL using the STAR aligner v.2.5.2b). BAM files were generated as a result of this step. Unique gene hit counts were calculated by using feature Counts from the Subread package v.1.5.2. Only unique reads that fell within exon regions were counted. After extraction of gene hit counts, the gene hit counts table was used for downstream differential expression analysis. Using DESeq2, a comparison of gene expression between the groups of samples was performed. The Wald test was used to generate P values and Log2 fold changes. Genes with adjusted P values < 0.05 and absolute log2 fold changes >1 were called as differentially expressed genes for each comparison. Gene ontology analysis was performed on the statistically significant set of genes by implementing the software GeneSCF. The goa Mus musculus GO list was used to cluster the set of genes based on their biological process and determine their statistical significance. A PCA analysis was performed using the "plotPCA" function within the DESeq2 R package. The plot shows the samples in a 2D plane spanned by their first two principal components. The top 500 genes, selected by highest row variance, were used to generate the plot. The bulk HUVEC RNA-seq data obtained in this publication have been deposited in NCBI's Gene Expression Omnibus and are accessible through GEO Series accession number GSE244230.

## Statistics

Statistical differences among the indicated groups were analyzed by unpaired two tailed Student's *t*-test or one-way or two-way Analysis of Variance (ANOVA) using Tukey's multiple comparison tests. All statistical analyses were done using Graph Pad Prism 9 software. A p value of <0.05 was considered significant. The *n* numbers and other significant values are indicated in the figures and figure legends (*: p<0.05; **: p<0.01; ***: p<0.001; ****:p<0.0001; ns-non significant).

## Supporting information

**S1 Fig. Accumulation of neutrophils and other immune cells in murine lung post Mtb infection. (A)** The gating strategy used to identify myeloid cells from total lung single-cell suspensions in BL/6 mouse lungs, delineating Neutrophils: Live CD11b+ Ly6G+; Monocytes: Live CD11b+ Ly6G- Ly6C+; Alveolar Macrophages: Live CD11b+ Ly6G- F4/80+ Siglec F+; Interstitial Macrophages: Live CD11b+ Ly6G- F4/80+ Siglec F-; Eosinophils: Live CD11b + Ly6G- Ly6C- Siglec F+; Plasmacytoid Dendritic cells (pDC): Live CD11b- Ly6G- Ly6C + Siglec H+. **(B)** Enumeration of monocytes, alveolar macrophages, interstitial macrophages, eosinophils, and plasmacytoid dendritic cells in the lungs of resistant BL/6 and susceptible C3H mice at day 29 pi. **(C)** Counts of infected monocytes, alveolar macrophages, interstitial macrophages, eosinophils, and plasmacytoid dendritic cells in BL/6 and C3H mice at 29 dpi. **(D)** 6-8-week-old BL/6, C3H, inos$^{-/-}$, and *Il1r1*$^{-/-}$ mice were infected with Mtb HN878 *smyc*:: mCherry SSB-GFP and evaluated at 29 dpi. Total live neutrophils (Live CD11b+ Ly6G+) numbers are shown for each mouse strain. Infected neutrophils (Live CD11b+ Ly6G+ *smyc*':: mCherry+) counts are presented for each mouse group. Sample sizes were n = 3–7 mice per group, representative of 2 experiments. Error bars represent Mean ± SD. Statistical analyses were performed using two-way ANOVA for (B, C) and ordinary one-way ANOVA for (D, E). Tukey's multiple comparison tests were used to calculate statistical significance. Significance is indicated as *p<0.05, **p<0.01, ***p<0.001, ****p<0.0001; ns indicates non-significant differences. Clip art/Images within figure panels were created with www.BioRender.com. (TIF)

**S2 Fig. Cellular dynamics in rabbit lungs infected with Mtb. (A)** Gating strategy for sorting neutrophils from C3H mouse lungs at 29 dpi: CD11b+ Ly6G+, prepared for confocal microscopy as shown in Fig 1F. **(B)** Description of the gating strategy for isolating CD11b- cells, alveolar macrophages (CD11b+ CD64$^{lo}$, Siglec F+), and monocytes (CD11b+ CD64+, Siglec F-) from C3H mouse lungs at day 29 pi. **(C)** Representative confocal microscopy images of the sorted cell types after cytospin preparation. CD64 is visualized with Cy5, *smyc*'::mCherry + identifies Mtb bacilli, and SSB-GFP foci are highlighted, with DAPI used as a counterstain. The top panel depicts CD11b- cells without CD64 or bacterial presence. The middle panel shows alveolar macrophages characterized by low CD64 levels and intracellular Mtb. The bottom panel displays monocytes with CD64 and absence of bacteria. Cell quantification from three mice is provided, indicating the percentage of cells harboring bacteria from CD11b-, AM, Monocytes and Neutrophils (Right). **(D)** Analysis of the bacterial burden in rabbit lungs infected with strains Mtb HN878 and CDC1551 at 2- and 4-weeks pi, using CFU counts. **(E)** Immunohistochemical evaluation of rabbit lung tissue infected with Mtb HN878 (left panel) and Mtb CDC1551 (right panel) at 2- and 4-weeks pi. Enlarged areas are indicated by square boxes in the left panels, with arrows in the right panels highlighting neutrophils, areas of necrosis, and lymphocytes. **(F)** Acid-fast staining of Mtb HN878 in infected rabbit lungs at 2- and 4-weeks pi, indicating bacterial presence and distribution. Sample sizes were n = 3 mice per group; 3 fields of view per mouse for (C) and n = 4 rabbits per group for (D). Error bars represent Mean ± SD. Statistical analyses were performed with ordinary one-way ANOVA for (C) and two-way ANOVA for (D). Tukey's multiple comparison tests were applied to determine statistical significance. Significance levels are indicated: *p<0.05, **p<0.01, ***p<0.001, ****p<0.0001; ns denotes a non-significant result. Clip art/Images within figure panels were created with www.BioRender.com.
(TIF)

**S3 Fig. Enhanced expression of metabolic pathway genes in the host response to Mtb infection. (A)** Diagram illustrating genes associated with fatty acid metabolism. Highlighted in red are genes expressed in neutrophils from mice infected with Mtb HN878. **(B)** Heatmaps of Gene Expression: Differential gene expression related to glycolysis and β-oxidation pathways in rabbit lungs at 2- and 4-weeks post-infection with Mtb CDC1551 and Mtb HN878, as determined by genome-wide microarray analysis; (n = 4 rabbits/ group). **(C)** Fatty Acid Oxidation Gene Expression: Graphs depict log2 expression counts for selected genes involved in the fatty acid oxidation pathway. **(D)** Glycolysis Pathway Gene Expression: Highlighting the expression levels of genes involved in glycolysis. (C, D) Analyses based on the GSE94438 dataset from RNA sequencing of whole blood cells from TB patients and healthy controls in a household contact study. Error bars represent Mean ± SD. Statistical significance in (C) and (D) was assessed using an unpaired t-test, with p-values denoted as follows: *p<0.05, **p<0.01, ***p<0.001, ****p<0.0001; ns indicates a non-significant result. Illustrations created with www.BioRender.com.
(TIF)

**S4 Fig. Metabolic substrate utilization by neutrophils during Mtb infection. (A)** Bone marrow neutrophils from 6–8-week-old BL/6 mice were isolated and infected *ex-vivo* with Mtb HN878 *s-myc*'::mCherry at an MOI of 3. After washing to remove extracellular bacteria 4 hours pi, cells were incubated for another 20 hours. The metabolic activity of neutrophils post-infection is assessed using fluorescent glucose and fatty acid analogs. Both infected and uninfected neutrophils were then labeled with C16-BODIPY or 2-NBDG 4 hours before harvesting and analyzed via flow cytometry. **(B)** 2-NBDG Uptake: Flow cytometry histogram and MFI

graph comparing 2-NBDG uptake (15μM) in uninfected versus infected neutrophils, indicating changes in glucose metabolism post-infection. **(C)** C16-BODIPY Labeling: Histogram and MFI graph showing fatty acid analog C16-BODIPY labeling (25μM) in uninfected versus infected neutrophils, reflecting fatty acid metabolism alterations upon infection. **(D)** Flow cytometry histogram and MFI graph of 2-NBDG in neutrophils with and without bacteria following *ex-vivo* challenge with Mtb HN878. **(E)** Histogram and MFI graph of C16-BODIPY in Mtb+ (infected) and Mtb- (uninfected) neutrophils upon *ex-vivo* challenge with Mtb HN878. Bone marrow neutrophils from BL/6 mice were infected with Mtb HN878 *smyc'*::mCherry, SSB-GFP at MOI-3. The cells were washed to remove extracellular bacteria 4 hours pi and were analyzed 24 hours pi. They were treated with increasing concentrations of isoniazid (INH) and MFI of mCherry **(F)** and neutrophil bacterial burden by CFU was enumerated **(G).** Sample size n = 3 replicates per group. Error bars denote Mean ± SD. Statistical analyses performed include ordinary one-way ANOVA for (B, C, F, G), with subsequent Tukey's multiple comparison tests, and unpaired t-tests for (D, E). Significance is denoted as follows: *p<0.05, **p<0.01, ***p<0.001, ****p<0.0001; ns indicates a non-significant difference. Illustrations created with www.BioRender.com.
(TIF)

**S5 Fig. Impact of fatty acid metabolism inhibitors on cell viability and antimycobacterial activity. (A)** The cytotoxicity of neutrophils treated with fatty acid oxidation inhibitors at indicated concentrations (Etomoxir, Ranolazine, Mildronate, and TMZ) was assessed at 24 hours pi by flow cytometry, reporting the percentage of cells that are fixable viability dye (FVD) negative. **(B)** Neutrophils were treated with either 0.1mM, 0.5mM, or 5mM 2-Deoxy Glucose (2DG), followed by measurement of bacterial burden by CFU counts at 24 hours pi. **(C)** Bone marrow neutrophils from C3H mice were infected with Mtb HN878 *smyc'*::mCherry, SSB-GFP *ex-vivo*. Cells were washed 4 hours pi to remove extracellular bacteria and were incubated further for 20 hours. Neutrophils were treated with 200μM ETO/ 100μM Ran/ 100μM Mil/ 100μM TMZ/ 0.5mM 2-DG and 24 hours pi, MFI of mCherry was assessed. **(D)** Bacterial burden in these neutrophils from (C) were enumerated by CFU counts. **(E)** CFU graph depicting the inhibitory effects of various concentrations of metabolic inhibitors, including 500μM Etomoxir, 100μM Mildronate, 100μM TMZ, 500μM Ranolazine, and 5mM 2-DG on Mtb broth culture. **(F)** Flow cytometry analysis of neutrophil cell death post treatment with mitochondrial metabolism inhibitors (Etomoxir, Oxfenicine, UK5099, R162, as used in **Fig 3F**) or with Etomoxir after supplementation with the medium-chain fatty acid, Octanoic acid, at 24 hours pi (% FVD- cells). **(G)** Bone Marrow Derived Macrophages (BMDM) were harvested from naïve BL/6 mice and infected with an MOI-3 of Mtb HN878 reporter. Graphs displaying the MFI of Mtb HN878 *smyc'*::mCherry at 3 and 5 dpi, compared to untreated controls. **(H)** Bacterial load in BMDMs at various time points post infection was determined by CFU counts. The experiments were conducted with n = 3 replicates per group, representative of two experiments. Error bars indicate Mean ± SD. Statistical analyses were performed using one-way ANOVA for (A-G), and two-way ANOVA for (H). Tukey's multiple comparison tests were used post hoc to determine significance: *p<0.05; **p<0.01; ***p<0.001; ****p<0.0001; ns denotes non-significant results. Clip art/Images within figure panels were created with www. BioRender.com.
(TIF)

**S6 Fig. Generation of a neutrophil specific knockout of *Cpt1a*. (A)** Infographic showing the generation of neutrophil-specific *Cpt1a* knockout mice (Mrp8-cre Cpt1a^fl/fl, designated as Cpt1a^Δneut). **(B)** Genotyping to validate the presence of MRP8-Cre (left) and Cpt1a loxP (right) genes in Cpt1a^Δneut mice. **(C)** Western blot analysis to confirm the absence of Cpt1a

(Size: ~86 kDa) in Mrp8 Cre+ Cpt1a $^{fl/fl}$ mouse bone marrow neutrophils, isolated by magnetic sorting and presence of Cpt1a in Mrp8 Cre- Cpt1a loxP neutrophils and other bone marrow cells in Cre+ and Cre- mice (top panel) with housekeeping gene Ubiquitin (bottom panel). Illustrations created with www.BioRender.com.
(TIF)

**S7 Fig. Reduced neutrophil infiltration is seen in fatty acid oxidation inhibited C3H mice post Mtb infection. (A)** Confocal microscopy images of lung sections (top to bottom) from control, etomoxir, ranolazine, mildronate, TMZ, and 2-DG treated mice (from **Fig 4A**). Mtb is visualized using *smyc*'::mCherry, SSB foci are visualized by GFP, neutrophils are stained with Ly6G-Cy5, and nuclei are counterstained with DAPI. **(B)** Total cell fluorescence of Ly6G staining was quantified by Image J. Representative image of n = 3 mice / group, 3 fields of view per mouse, 200–500 bacteria per image. Error bars indicate Mean ± SD; Scale- 5μm. Statistical analysis was performed using one-way ANOVA (B) with Tukey's multiple comparison test for significance: $p < 0.05$, *$p < 0.01$, **$p < 0.001$, ***$p < 0.0001$; ns, not significant.
(TIF)

**S8 Fig. Effect of fatty acid metabolism inhibition on inflammatory cell recruitment in TB diseased lungs. (A)** Representative flow cytometric plots display the Ly6G$^{hi}$ and Ly6G$^{lo/dim}$ neutrophil profiles from BL/6 and C3H untreated control, 2-DG-treated, and Etomoxir-treated mice on 29 dpi with Mtb HN878 *smyc*'::mCherry, SSB-GFP. **(B)** Graphs depict the counts of various live cell types, including monocytes (Live CD11b+ Ly6G- Ly6C+), macrophages (Live CD11b+ Ly6G- Ly6C- F4/80+), eosinophils (Live CD11b+ Ly6G- Ly6C- F4/80- SigF+), and dendritic cells (Live CD11b- Ly6G- Ly6C+ SigH+), in the lungs of untreated control, 2-DG-treated, and Etomoxir-treated mice at 29 dpi. **(C)** Total numbers of infected immune cells (*smyc*'::mCherry+), including monocytes, macrophages, eosinophils, and dendritic cells, are compared among Control, 2-DG, and Etomoxir-treated mice at 29 dpi. **(D)** The percentage of live CD4+ T-cells (Live CD11b- CD19- CD3+ CD4+), CD8+ T-cells (Live CD11b- CD19- CD3+ CD8+), and CD19+ B-cells (Live CD11b- CD3- CD19+) within the lungs of untreated controls, 2-DG, and Etomoxir-treated mice are shown at 29 dpi. Sample size n = 6 mice per group, representative data from two experiments. For (H), n = 3 mice/ group, in triplicates. Error bars indicate Mean ± SD. Statistical analysis was performed using one-way ANOVA with Tukey's multiple comparison tests for significance: *$p<0.05$, **$p<0.01$, ***$p<0.001$, ****$p<0.0001$; ns denotes non-significant differences. Clip art/Images within figure panels were created with www.BioRender.com.
(TIF)

**S9 Fig. Downregulation of inflammatory cytokine release by fatty acid metabolism inhibition in Mtb Infected Mice.** Cytokine levels were quantitatively analyzed in lung homogenates from mice treated with 2-DG or Etomoxir in comparison to untreated controls. The graphs present a comparison of various cytokines, including Eotaxin, IL-2 to IL-17, IP-10, KC, LIF, LIX, MCSF, MCP-1, MIG, MIP-2, RANTES, VEGF, and GM-CSF, measured in picograms per milliliter (pg/ml). Sample size n = 3 mice per group, in triplicates. Error bars represent Mean ± standard deviation (SD). Statistical analyses were conducted using one-way ANOVA with Tukey's multiple comparison tests to determine significance. The stars indicate levels of statistical significance with *$p<0.05$, **$p<0.01$, ***$p<0.001$, ****$p<0.0001$; 'ns' denotes non-significant differences.
(TIF)

**S10 Fig. Immune cell recruitment to the lungs of Cpt1a$^{Δneut}$ mice and littermate control mice post TB disease. (A)** Enumeration of total numbers of live monocytes (Live CD11b

+ Ly6G- Ly6C+), macrophages (Live CD11b+ Ly6G- Ly6C- F4/80+), eosinophils (Live CD11b + Ly6G- Ly6C- F4/80- SigF+), and dendritic cells (Live CD11b- Ly6G- Ly6C+ SigH+), in the lungs of littermate controls and Cpt1a$^{\Delta neut}$ mice (left) and α-IL1R-treated controls and Cpt1a$^{\Delta neut}$ mice (right) infected with Mtb HN878 *smyc':*:mCherry at 29 dpi. **(B)** Quantification of the total number of infected (*smyc':*:mCherry+) monocytes, macrophages, eosinophils, and dendritic cells in the lungs of control and Cpt1a$^{\Delta neut}$ mice (left) and α-IL1R-treated control and Cpt1a$^{\Delta neut}$ mice (right) at 29 dpi. **(C)** Proportion of total CD3+ T-cells (Live CD11b- CD19- CD3+) and **(D)** CD19+ B-cells (Live CD11b- CD3- CD19+) in control and Cpt1a$^{\Delta neut}$ mice (left) and α-IL1R-treated control and Cpt1a$^{\Delta neut}$ mice (right) at 29 dpi. Sample size n = 4 mice per group. Error bars indicate Mean ± SD, are from one experiment. Statistical analysis was performed using unpaired t-tests for significance: $p < 0.05$, *$p < 0.01$, **$p < 0.001$, ***$p < 0.0001$; ns, not significant.
(TIF)

**S1 Table. Quantification of Ly6G intensity in Mtb infected mouse lungs.** 1 field of view each from Mtb infected BL/6 and C3H mouse lungs were randomly chosen to show the method of quantification of Ly6G staining intensity.
(XLSX)

**S2 Table. RAW data used to generate the figures of this manuscript.**
(XLSX)

## Acknowledgments

We are grateful for the assistance of Christina Peterson, Nusret Bekir Subasi and Rebecca Pirri of Pathology and Laboratory Medicine core of Albany Medical College for their help with histopathology and scanning lung tissues. We thank Dr. Karen Krause, Cindy Vanvorst, Victoria Boppert and Vicente Baz of Animal Research Facility, Dr. Amit Singh in the BSL-3 management of Albany Medical College for their unwavering support with animal husbandry and care of laboratory animals used in this study and Dr. Don Steiner in the IMD Core Facility for help with cell sorting and maintenance of the flow cytometer. Authors acknowledge Advanced Light Microscopy and Imaging Analysis core facility of Wadsworth Center, NYSDOH.

## Author Contributions

**Conceptualization:** Poornima Sankar, Bibhuti B. Mishra.

**Data curation:** Poornima Sankar, Ramon Bossardi Ramos, Jamie Corro, Lokesh K. Mishra, Tanvir Noor Nafiz, Gunapati Bhargavi, Mohd Saqib, Sibongiseni K. L. Poswayo, Bibhuti B. Mishra.

**Formal analysis:** Poornima Sankar, Bibhuti B. Mishra.

**Funding acquisition:** Bibhuti B. Mishra.

**Investigation:** Bibhuti B. Mishra.

**Resources:** Ramon Bossardi Ramos, Suraj P. Parihar, Yi Cai, Selvakumar Subbian, Anil K. Ojha, Bibhuti B. Mishra.

**Supervision:** Bibhuti B. Mishra.

**Writing – original draft:** Poornima Sankar, Bibhuti B. Mishra.

**Writing – review & editing:** Bibhuti B. Mishra.

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
