## [Decision Letter · Decision Letter 0]

28 May 2024

Dear Dr. Mishra,

Thank you very much for submitting your manuscript "Fatty acid metabolism in neutrophils promotes lung damage and bacterial replication during tuberculosis." for consideration at PLOS Pathogens. As with all papers reviewed by the journal, your manuscript was reviewed by members of the editorial board and by several independent reviewers. In light of the reviews (below this email), we would like to invite the resubmission of a significantly-revised version that takes into account the reviewers' comments.

The revised manuscript offers important insights into the role of neutrophil metabolism in controlling Mtb infection. Specifically, authors try to establish a direct link between glycolysis as opposed to fatty acid oxidation in neutrophils and control of infection in vivo. The interpretation of data relating to the importance of neutrophils in being key mediators of infection control in vivo is complicated by the global effects of the metabolic inhibitors on other cell types where these inhibitors have been reported by others to mediate infection control in, for example, macrophages. The demonstration that the Cpt1a knockout in neutrophils in mice impacts disease burden or pathology would be the most ideal demonstration of the role of neutrophil metabolism in regulating Mtb infection. The authors should also put more emphasis in the discussion on putting their work in the context of previous work in the field that explores the role of glycolysis and fatty acid oxidation in Mtb disease progression. The authors need to repeat key ex vivo experiments with neutrophils obtained from C3H mice given the important findings concerning differences in neutrophil data generated in this genetic background compared to B6 mice. Other important and technically less challenging experiments to include are addressing concerns regarding details regarding quantification of replication status and cell type, better assay description, the need to show cytometry results from in vitro Mtb with defined growth/ replication states, confirm the absence of Cpt1a expression (qRT-PCR or Western blot) and repeat the Western blot in Fig. 2G.

We cannot make any decision about publication until we have seen the revised manuscript and your response to the reviewers' comments. Your revised manuscript is also likely to be sent to reviewers for further evaluation.

Sincerely,

Helena Ingrid Boshoff

Section Editor

PLOS Pathogens

Helena Boshoff

Section Editor

PLOS Pathogens

Michael Malim

Editor-in-Chief

PLOS Pathogens

orcid.org/0000-0002-7699-2064

The revised manuscript offers important insights into the role of neutrophil metabolism in controlling Mtb infection. Specifically, authors try to establish a direct link between glycolysis as opposed to fatty acid oxidation in neutrophils and control of infection in vivo. The interpretation of data relating to the importance of neutrophils in being key mediators of infection control in vivo is complicated by the global effects of the metabolic inhibitors on other cell types where these inhibitors have been reported by others to mediate infection control in, for example, macrophages. The demonstration that the Cpt1a knockout in neutrophils in mice impacts disease burden or pathology would be the most ideal demonstration of the role of neutrophil metabolism in regulating Mtb infection. The authors should also put more emphasis in the discussion on putting their work in the context of previous work in the field that explores the role of glycolysis and fatty acid oxidation in Mtb disease progression. The authors need to repeat key ex vivo experiments with neutrophils obtained from C3H mice given the important findings concerning differences in neutrophil data generated in this genetic background compared to B6 mice. Other important and technically less challenging experiments to include are addressing concerns regarding details regarding quantification of replication status and cell type, better assay description, the need to show cytometry results from in vitro Mtb with defined growth/ replication states, confirm the absence of Cpt1a expression (qRT-PCR or Western blot) and repeat the Western blot in Fig. 2G.

Reviewer's Responses to Questions

**Part I - Summary**

Reviewer #1: In this manuscript, Sankar et al examine the role of neutrophils during Mtb infection, focusing on the impact of host cell metabolism on infection outcome. In particular, using various inhibitors as well as a neutrophil-specific Cpt1a knockout mouse, they report that inhibition of fatty acid oxidation decreases Mtb survival in neutrophils. The authors present extensive data, and the findings are overall interesting. However, I have several questions regarding some experiments and conclusions drawn, as detailed below.

Reviewer #2: This manuscript focused on the metabolism of neutrophils during Mtb infection is significant as this topic is much less well studied compared to macrophage metabolism during Mtb infection.

Reviewer #3: In this manuscript by Sankar et al the authors examine the mechanisms driving neutrophil-mediated TB pathogenesis in susceptible animals. Using the clinical isolate HN878 the authors show neutrophils are associated with poor disease outcomes and a key site of bacterial replication in C3H (Kramnick) mice (and other susceptible animals like Nos2) as well as rabbits. To understand how differences in neutrophils drive disease RNAseq analysis was completed in C3H and B6 mice identifying an increase in metabolic genes related glycolysis and fatty acid oxidation (FAO). FAO was also identified as differentially induced in both rabbits and humans suggesting a possible link. Using an ex vivo neutrophil model from B6 mice the authors then show that blocking FAO in neutrophils drives more Mtb restriction using a combination of genetic and chemical approaches. Using these inhibitors in vivo the authors then show that FAO is globally important to drive Mtb disease in C3H mice. Finally the authors show that FAO is important for neutrophils to migrate towards infected macrophages and may play a role in NET formation as a mechanism of Mtb killing. This paper is generally well written and there is a great deal of experiments across distinct model systems that is a key strength in addition to using a the clinical isolate HN878 with nice bacterial replication reporters. While many studies, including some by this investigator, have shown that neutrophils are a driving factor of intracellular Mtb growth and disease progression in susceptible animals, the identification of distinct metabolic networks in neutrophils that may contribute to this observation is important for the field. However, there are some concerns about the experimental design in several figures that may limit the conclusions that are strongly made by the investigator. Overall, this study is likely to interesting to both TB investigators and those studying the role of neutrophils in disease but some shortcomings in the overall approach lower my overall enthusiasm.

**Part II – Major Issues: Key Experiments Required for Acceptance**

Reviewer #1: - How were the SSB-GFP foci quantified for the graphs shown in Fig. 1C and Fig. 4E? In particular, is one data point from one mouse? It is unclear from the figure legend or the methods how this quantification was conducted. Particularly given the range in the percentage of Mtb that exhibit SSB-GFP foci that can be observed across images, robust quantification requires enumeration of a percentage of SSB-GFP foci from observations across multiple images for one mouse, with one mouse thus providing one data point. Can the authors please clarify.

- Related to the above, it is overall unclear how the determination regarding Mtb replication state was made for the various host cell types (lines 145-156). Were these conclusions made from microscopy quantification (and if so, how exactly was this done; see questions above), or from flow cytometry (Fig. S2A)? The flow cytometry plot in Fig. S2A does not robustly show separation of SSB-GFP positive versus negative Mtb. And indeed the one bacterium shown as an example for each class in the cytospin in Fig. 1F raise concerns regarding the ability of flow cytometry to properly distinguish SSB-GFP positive versus negative Mtb – the one Mtb bacillus shown as "mCherry+ GFP lo" actually possesses an SSB-GFP focus, and should be considered as being in a replicating state. SSB-GFP signal is present to some degree in all bacteria regardless of DNA replication status, with the localization of signal that differs when DNA replication is active, as the authors note. A control flow cytometry sorting using Mtb in known states of replication versus not (i.e. from in vitro Mtb experiments in defined conditions), with appropriate microscopy comparison, is needed to establish flow cytometry as a means for successful separation of SSB-GFP positive versus negative Mtb. In the absence of robust establishment that flow cytometry is able to properly distinguish SSB-GFP-positive from SSB-GFP-negative Mtb, analysis should be conducted via microscopy only, with appropriate numbers of images/animals, as noted above.

- The exacerbation of Mtb infection with 2-DG treatment, versus the improved control of Mtb infection by FAO inhibition, has previously been shown by various groups. In this context, the effect of these treatments on other key immune cell types for Mtb (e.g. alveolar versus interstitial macrophages) have also been described. It is important that the authors acknowledge prior literature on this and incorporate that information into their conclusions and discussion. Indeed, as the authors note, "ETO administration reduced the infiltration of monocytes, macrophages, eosinophils, and plasmacytoid dendritic cells (Fig. S6B), with a significant decrease in the number of infected alveolar and interstitial macrophages (Fig. S6C)". How do the authors differentiate the relative contributions of the alterations to the different host immune cell types to the in vivo outcome with Mtb infection observed?

- Fig. 5A, lines 383-384: The authors state that there is a "dose-dependent inhibition of neutrophil migration by ETO", but the data shown do not in fact demonstrate any dose-dependence. This should be clarified and the conclusions adjusted accordingly.

- The materials and methods and the figure legends would benefit from added details, as it is currently difficult in several instances for a reader to figure out what is being shown/how an assay/quantification was performed. To list a few examples: It is not clear how quantification in Fig. 1G was performed. How are differences in total number of cells in an image accounted for? What does one data point in the graph represent? One mouse? For all graphs, it should be made clear what data is being presented. In Fig. 2G, the figure currently only says "WB". Unclear why the sentence after states sample size for RNA sequencing? In Fig. S2D, information regarding how many animals were in each infection set/time point needs to be added. In Fig. 4D, how was "% necrotic area" determined? How many animals were in each treatment set and how many images/animal were quantified? In Fig. 4E, no key is provided for what the different colors in the images represent. In Fig. 5A, the main text and figure legend state incubation was for 20 hours, while the figure itself states 12 hours.

Reviewer #2: 1. The entire manuscript is largely relied on various inhibitors and in vitro studies. In the revision, authors have generated a mouse model that specifically lack Cpt1a (a key enzyme regulating fatty acid oxidation) in neutrophils. This is a great genetic approach to test their hypothesis. However, authors only utilized bone marrow neutrophils isolated from these mice for in vitro studies. Although the data are in line with other experiments utilizing inhibitors, it would be essential and critical for them to test whether fatty acid metabolism in neutrophils promotes Mtb infection in vivo using this mouse model. This experiment will be highly valuable. The authors have the resource to perform this experiment.

2. Regarding the mouse model that I mentioned above, in Figure 3G, they only showed the genotyping results. qPCR or western blot data are lacking to confirm Cpt1a is not expressed in neutrophils in those mice.

3. In Figure 2G, the western blot showing here is concerning, given the level of b-actin in Mtb-infected neutrophils at 24h is significantly less compared to other samples. This implicates that the amounts of protein loaded to the gel are very different. It is also possible that the expression of b-actin in neutrophils was altered after Mtb infection at 24hours. If this is the case, other housekeeping genes will be needed to use as the loading control. The figure legend of this panel is over-simplified without any detailed information.

Reviewer #3: Major Comments:

1. There is a disconnect in some of the logic through the paper that should be addressed to better support the authors conclusions. First, the RNAseq analysis from neutrophils in B6 v C3H found that FAO was increased in C3H but not B6. No real metabolic analysis was done on these cells and RNA expression is not always indicative of actual metabolic output. Furthermore, it seems all neutrophils were analyzed so some discussion on how these transcriptional/metabolic changes differ between infected and uninfected neutrophils should be at the very least addressed as a caveat (ie do these neutrophils NEED to be infected to drive this metabolic shift?). Along these lines it is a little surprising that that all ex vivo work is done in B6 neutrophils that did not show increased FAO and not C3H neutrophils. This seems like a missed opportunity to directly validate their in vivo studies ex vivo in a comparative way that would more fully support the conclusions.

2. There are key concerns about the in vivo inhibitor experiments that globally block FAO and glycolysis in vivo. Given the importance of these metabolic pathways in a range of cell types Mtb interacts with, some shown here by the authors and previously published by others, it is hard to conclude from this experimental setup that blocking these metabolic pathways is specifically due to changes in neutrophils. It could be these inhibitors are altering Type I IFN and other cytokines in the macrophages that then alter neutrophils recruitment and function during infection. Given the authors made a neutrophil specific CPT1 mouse it seems leveraging this in one of the other susceptible mouse models (ie Nos2 or IL1R) or leveraging recently published SP140 mice that phenocopy many aspects of C3H mice on the B6 background are really needed to support the authors conclusions. I understand that this is a large undertaking, but if the authors really want to make the conclusions that neutrophil specific FAO is critical in vivo as written it seems to be very important to not globally block these metabolic pathways which may have many pleotropic effects on infection and disease progression.

3. The transition of the final figure of the manuscript to focus on NETs feels disjointed and a little underdeveloped. Cybb is known to be a key player in driving NET formation (one example: PMID: 17210947). How do the authors explain that their phenotype is not observed in Cybb neutrophils then? This suggests there may be something else distinct between the Pad4 and Cybb neutrophils that is not dependent on NETS. However, without directly quantifying whether NETs are produced, instead of relying on just microbial readouts, it is difficult to interpret and make strong conclusions from these data. I am not convinced these findings are needed given their preliminary state and I may cautiously suggest removing them to keep the focus of the manuscript clearer and more supported by the data. Alternatively, more experiments to support these conclusions and clarify the results are needed.

**Part III – Minor Issues: Editorial and Data Presentation Modifications**

Reviewer #1: - 2-NBDG and BODIPY have overlapping spectra with GFP. Since the bacteria used in the experiments shown in Fig. S4 carry the SSB-GFP reporter, is it the case that the 2-NBDG and BODIPY signal were much stronger and essentially the settings were such that SSB-GFP signal was not detected?

- Line 261: It would not appear that there is any Mtb growth observed in the 24 hour infection of neutrophils ex vivo, given the starting inoculum of 6 x 10^6 Mtb/well stated in the methods versus the ~1.25-2.25 x 10^6 Mtb/well obtained from mock-treated cells in Fig. 3B. This sentence should be rephrased. Is there a reason why Fig. 3B data is shown in a different graph format than Figs. 3C-3E?

- Fig. 3F: Octanoic acid alone improves Mtb growth, which makes it somewhat difficult to draw conclusions regarding the phenotype observed with ETO + octanoic acid. At minimum the authors’ conclusion regarding the use of octanoic acid to mitigate the effects of ETO should be tempered.

- Are there other effects of Pad4 deletion beyond NET formation?

- "smyc'" should be italicized.

Reviewer #2: The statistical information is lacking in Figure 4D. It looks like only one sample from each group was used to calculate % necrotic lesion area.

Reviewer #3: Minor Comments:

1. It was not described why for the transwell experiment human macrophages were used as a source of infected cells while mouse neutrophils were then used. It seems some differences between mouse and human could confound these results and it is straightforward to use murine BMDMs or other macrophage lines to directly answer this.

2. While I understand that ex vivo neutrophils are challenging given how rapidly the die, I do have some concerns around the robustness of the mCherry/CFU experiments in neutrophils given the very short timescale. First, mCherry has a very stable half life thus seeing a reliable decrease in MFI (even if modest) may be difficult to make reliable conclusions from. Including an antibiotic treatment control in these experiments where you know Mtb is being killed would help alleviate some of these concerns given the short time scale. Alternatively using imaging approaches with the SSB reporter to more quantitatively assess bacterial burden in these cells may be another approach to better support these conclusions.

PLOS authors have the option to publish the peer review history of their article (what does this mean?). If published, this will include your full peer review and any attached files.

Reviewer #1: No

Reviewer #2: No

Reviewer #3: No
---

## [Decision Letter · Decision Letter 1]

9 Sep 2024

Dear Dr. Mishra,

Thank you very much for submitting your manuscript "Fatty acid metabolism in neutrophils promotes lung damage and bacterial replication during tuberculosis." for consideration at PLOS Pathogens. As with all papers reviewed by the journal, your manuscript was reviewed by members of the editorial board and by several independent reviewers. The reviewers appreciated the attention to an important topic. Based on the reviews, we are likely to accept this manuscript for publication, providing that you modify the manuscript according to the review recommendations.

Please address the comments of reviewer 1.

Sincerely,

Helena Ingrid Boshoff

Section Editor

PLOS Pathogens

Helena Boshoff

Section Editor

PLOS Pathogens

Michael Malim

Editor-in-Chief

PLOS Pathogens

orcid.org/0000-0002-7699-2064

Please address the comments of reviewer 1.

Reviewer Comments (if any, and for reference):

Reviewer's Responses to Questions

**Part I - Summary**

Reviewer #1: The authors have addressed most points of concern in this revised manuscript, and I have just a few remaining questions.

Reviewer #2: The authors have adequately addressed all my concerns. I am happy to accept this manuscript for publication.

Reviewer #3: This revised manuscript examine how Fatty Acid metabolism in neutrophils contributes to disease severity during Mycobacterium tuberculosis. The authors have added significantly more data with their Cpt KO mouse, as well as addressed lipid metabolism in a susceptible mouse strain. The added data help to support the conclusions more directly and these data will be of great interest to immunologists and mycobacterial researchers.

two small changes may further help set the context and support the conclusions :

1. It is unclear from the figure legends how many times each experiment was completed - especially with the new in vivo experiments. This should be made more explicit for readers assess the reproducibility of the results.

2. A recent study published in mucosal immunology supports some of the finding here (PMID: 38844208) - this study should at the very least be cited/mentioned/discussed as it provides additional support for the model proposed by the authors.

**Part II – Major Issues: Key Experiments Required for Acceptance**

Reviewer #1: - I do not see information regarding the SSB-GFP quantification (number of images and number of animals) added to the figure legends, even though that was stated as now included? (Needed information is noted as having been added to the methods.) There are many other instances where information is stated in the response to reviewers document as having been added to the manuscript, but are not actually present in the manuscript, with the referenced line numbers also correspondingly off. I am not sure if the wrong version of the revised manuscript was uploaded? All changes noted in the response to reviewer document should be carefully checked as indeed having been added as stated. In general, the figure legends also remain too sparse for effective understanding by readers without constant searching in the results/methods section for explanations. Just as one example, it is still not clear what the blot in Fig. 2G is showing. 2 different samples for -Mtb and 3 samples for +Mtb? From different experiments, or just repeat runs off the same experiment? Please edit throughout to improve ease of reader understanding.

- The anti-IL1R antibody treatment experiments are quite confusing as currently described, as while the neutrophil phenotypes track between +/- anti-IL1R treatment, other aspects do not. In Fig. S10B in particular, the Cpt1a∆neut mice have opposite phenotypes to their littermate controls depending on whether there is anti-IL1R treatment (i.e. in the absence of anti-IL1R treatment, there are higher numbers of infected monocytes, macrophages, eosinophils, and dendritic cells in the Cpt1a∆neut mice versus control, but the opposite is true in the presence of anti-IL1R treatment). How does that track with the decreased CFUs noted in Fig. 6B? At minimum, further explanation to put the results in better context is required for this set of experiments.

Reviewer #2: No major issues.

Reviewer #3: (No Response)

**Part III – Minor Issues: Editorial and Data Presentation Modifications**

Reviewer #1: - Lines 272-273: Something appears to be missing in the front part of the sentence - "Following 4-hour pi to remove extracellular bacteria,…" What was done?

- Line 355: "…demonstrated pivotal role of metabolism…" Add "the" or "a" in between "demonstrated" and "pivotal".

- Line 481: Remove "in" from "…while in mature neutrophils could…".

- Line 508: "which then passively transported…" Missing "is" between "which" and "then".

- Line 535: Add citations for the macrophage work.

- Line 541: "Suggesting", not "suggest".

- Fig. S10A is not currently ever referenced in the text.

Reviewer #2: No minor issues.

Reviewer #3: (No Response)

PLOS authors have the option to publish the peer review history of their article (what does this mean?). If published, this will include your full peer review and any attached files.

Reviewer #1: No

Reviewer #2: No

Reviewer #3: No

Figure Files:

Data Requirements:

Reproducibility:

References:

---

## [Editor Report · Decision Letter 2]

24 Sep 2024

Dear Dr. Mishra,

We are pleased to inform you that your manuscript 'Fatty acid metabolism in neutrophils promotes lung damage and bacterial replication during tuberculosis.' has been provisionally accepted for publication in PLOS Pathogens.

Best regards,

Helena Ingrid Boshoff

Section Editor

PLOS Pathogens

Helena Boshoff

Section Editor

PLOS Pathogens

Michael Malim

Editor-in-Chief

PLOS Pathogens

orcid.org/0000-0002-7699-2064

The authors have sufficiently addressed the reviewers' concerns
---

## [Editor Report · Acceptance letter]

1 Oct 2024

Dear Dr. Mishra,

We are delighted to inform you that your manuscript, "Fatty acid metabolism in neutrophils promotes lung damage and bacterial replication during tuberculosis.," has been formally accepted for publication in PLOS Pathogens.

Best regards,

Michael Malim

Editor-in-Chief

PLOS Pathogens

orcid.org/0000-0002-7699-2064